# AuG-KD: Anchor-Based Mixup Generation for Out-of-Domain Knowledge Distillation

**Zihao Tang, Zheqi Lv, Shengyu Zhang**[*]
Zhejiang University
{tangzihao,zheqilv,sy_zhang}@zju.edu.cn

**Yifan Zhou**
Shanghai Jiao Tong University
geniuszhouyifan@gmail.com

**Xinyu Duan**
Huawei Cloud
duanxinyu@huawei.com

**Fei Wu & Kun Kuang**[*]
Zhejiang University
{wufei,kunkuang}@zju.edu.cn

## Abstract

Due to privacy or patent concerns, a growing number of large models are released without granting access to their training data, making transferring their knowledge inefficient and problematic. In response, Data-Free Knowledge Distillation (DFKD) methods have emerged as direct solutions. However, simply adopting models derived from DFKD for real-world applications suffers significant performance degradation, due to the discrepancy between teachers' training data and real-world scenarios (student domain). The degradation stems from the portions of teachers' knowledge that are not applicable to the student domain. They are specific to the teacher domain and would undermine students' performance. Hence, selectively transferring teachers' appropriate knowledge becomes the primary challenge in DFKD. In this work, we propose a simple but effective method AuG-KD. It utilizes an uncertainty-guided and sample-specific anchor to align student-domain data with the teacher domain and leverages a generative method to progressively trade off the learning process between OOD knowledge distillation and domain-specific information learning via mixup learning. Extensive experiments in 3 datasets and 8 settings demonstrate the stability and superiority of our approach. Code available at https://github.com/IshiKura-a/AuG-KD

## 1 Introduction

With the surge of interest in deploying neural networks on resource-constrained edge devices, lightweight machine learning models have arisen. Prominent solutions include MobileNet (Howard et al., 2019), EfficientNet (Tan & Le, 2019), ShuffleNet (Ma et al., 2018), etc. Although these models have shown promising potential for edge devices, their performance still falls short of expectations. In contrast, larger models like ResNet (He et al., 2016) and CLIP (Radford et al., 2021), have achieved gratifying results in their respective fields (Wang et al., 2017; Tang et al., 2024). To further refine lightweight models' performance, it is natural to ask: can they inherit knowledge from larger models? The answer lies in Knowledge Distillation (Hinton et al., 2015) (KD).

Vanilla KD (Kim et al., 2023; Calderon et al., 2023) leverages massive training data to transfer knowledge from teacher models $T$ to students $S$, guiding $S$ in emulating $T$'s prediction distribution. Although these methods have shown remarkable results in datasets like ImageNet (Deng et al., 2009) and CIFAR10 (Krizhevsky, 2009), when training data is unavailable due to privacy concerns (Truong et al., 2021) or patent restrictions, these methods might become inapplicable.

To transfer $T$'s knowledge without its training data, a natural solution is to use synthesized data samples for compensation, which forms the core idea of Data-Free Knowledge Distillation (DFKD) (Binici et al., 2022; Li et al., 2023; Patel et al., 2023; Do et al., 2022; Wang et al., 2023a). These methods typically leverage $T$'s information, such as output logits, activation maps, intermediate outputs, etc., to train a generator to provide synthetic data from a normally distributed latent variable.

---

[*]Shengyu Zhang and Kun Kuang are corresponding authors.

The distillation process is executed with these synthesized data samples. However, DFKD methods follow the Independent and Identically Distributed Hypothesis (IID Hypothesis). They suppose that $T$'s training data (teacher domain $D_t$) and the real application (student domain $D_s$) share the same distribution (Fang et al., 2021b). In case the disparity between these two distributions cannot be neglected, these methods would suffer great performance degradation. Namely, the disparity is denoted as Domain Shift while the distillation without $T$'s training data under domain shift is denoted as Out-of-Domain Knowledge Distillation (OOD-KD). In Figure 1, we demonstrated the difference among KD, DFKD, and OOD-KD problems, where KD can access both $D_t$ and $D_s$, while DFKD can access neither $D_t$ or $D_s$. OOD-KD can access $D_s$, but has no prior knowledge of $D_s$. Moreover, KD and DFKD problems require the IID assumption between $D_t$ and $D_s$, which can hardly satisfied in real applications. Here, OOD-KD problem is designed for address the distribution shift between $D_t$ and $D_s$. Although domain shift has garnered widespread attention in other fields (Lv et al., 2023; 2024; Zhang et al., 2023c; Huang et al., 2021; Lv et al., 2022), there's no handy solution in OOD-KD (Fang et al., 2021a). MosiacKD (Fang et al., 2021a) is the state-or-the-art method for addressing OOD-KD problem, but it mainly focuses on the improvement of performance in $D_t$, ignoring the importance of $D_s$ (i.e. out-of-domain performance). Recently, some studies propose cross-domain distillation (Li et al., 2022; Yang et al., 2022) for OOD-KD, but these methods require grant access to $D_t$, which is impractical in real applications.

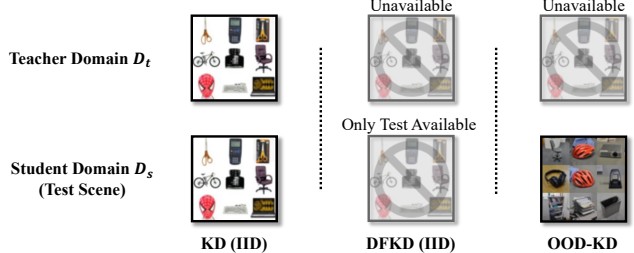

Figure 1: Differences between KD, DFKD, and OOD-KD problems.

In this paper, we focus on the problem of OOD-KD, and to address this problem we are still facing the following challenges: **(i) How to selectively transfer teachers' knowledge**. In OOD-KD problem, the difference of the joint distribution $P(X, Y)$ between teacher domain $D_t$ and student domain $D_s$ creates a significant barrier. Since $T$ is optimized for $D_t$, faced with data in $D_s$, $T$ is likely to give inaccurate predictions or fail to reflect the precise relationships between classes in $D_s$, impeding $S$'s performance unavoidably. **(ii) The absence of $T$'s training data makes OOD-KD extremely challenging**. As $T$'s training data act as the carrier of knowledge in vanilla KD, without it, knowledge transferring becomes troublesome. In contrast, data in the application scenes are easy to obtain. It is important to notice that their domain-specific information is applicable to $D_s$, if utilized properly, it is able to benefit $S$'s training.

To tackle these challenges, we propose a simple but effective method: Anchor-Based Mixup Generative Knowledge Distillation (AuG-KD). Our method utilizes an uncertainty-driven and sample-specific anchor to align student-domain data with $D_t$ and leverage a generative method to progressively evolve the learning process from OOD knowledge distillation to domain-specific information learning. Particularly, AuG-KD consists of 3 modules: Data-Free Learning Module, Anchor Learning Module, and Mixup Learning Module. Data-Free Learning Module bears semblance to vanilla DFKD, **tackling the absence of $D_t$**. Anchor Learning Module designs an uncertainty-aware AnchorNet to map student-domain samples to "anchor" samples in $D_t$, **enabling $T$ to provide proper knowledge for distillation**. Mixup Learning module utilizes the "anchor" samples to generate a series of images that evolve from $D_t$ to $D_s$, treating them as additional data for training. As the module progresses, $T$ becomes less certain about them while the domain-specific information gradually becomes important, **balancing OOD knowledge distillation and domain-specific information learning ultimately.** Extensive experiments attest to the excellent performance of our proposed method. In essence, our contributions can be briefly summarized as follows:

- We aim at an important and practical problem OOD-KD. To the best of our knowledge, we are the first to provide a practical solution to it.
- We propose a simple but effective method AuG-KD. AuG-KD devises a lightweight AnchorNet to discover a data-driven anchor that maps student-domain data to $D_t$. AuG-KD then adopts a novel uncertainty-aware learning strategy by mixup learning, which pro-

gressively loosens uncertainty constraints for a better tradeoff between OOD knowledge distillation and domain-specific information learning.
- Comprehensive experiments in 3 datasets and 8 settings are conducted to substantiate the stability and superiority of our method.

## 2 RELATED WORK

Since OOD-KD is a novel problem, we focus on the concept of Knowledge Distillation first. KD is a technique that aims to transfer knowledge from a large teacher model to an arbitrary student model, first proposed by Hinton et al. (2015). The vanilla KD methods either guide the student model to resemble the teacher's behavior on training data (Bucila et al., 2006) or utilize some intermediate representations of the teacher (Binici et al., 2022; Romero et al., 2015; Park et al., 2019). In recent years, knowledge distillation has witnessed the development of various branches, such as Adversarial Knowledge Distillation (Binici et al., 2022; Yang et al., 2023), Cross-Modal Knowledge Distillation (Li et al., 2022; Yang et al., 2022), and Data-Free Knowledge Distillation (Li et al., 2023; Patel et al., 2023; Do et al., 2022; Wang et al., 2023a).

Recently, data-free methods (DKFD) have garnered significant attention. DFKD typically relies on teacher models' information such as output logits and activation maps to train a generator for compensation from a normally distributed latent variable. Besides, there are also some sampling-based methods utilizing unlabeled data (Chen et al., 2021; Wang et al., 2023b). However, the effectiveness of DFKD methods is based on the assumption of the IID Hypothesis, which assumes that student-domain data is distributed identically to that in $D_t$. This assumption does not hold in many real-world applications (Arjovsky et al., 2019; Zhang et al., 2020; Liu et al., 2023), leading to significant performance degradation. The violation of the IID Hypothesis, also known as out-of-domain or domain shift, has been extensively discussed in various fields (Huang et al., 2021; Liang et al., 2022; Sagawa et al., 2020; Zhang et al., 2024b; 2023b; Qian et al., 2022). However, little attention has been paid to it within the context of knowledge distillation (Fang et al., 2021a). MosiacKD (Fang et al., 2021a) first proposes Out-of-Domain Knowledge Distillation but their objective is **fundamentally different from ours**. They use OOD data to assist source-data-free knowledge distillation and focus on **in-domain performance**. In contrast, we use OOD data for better **out-of-domain performance**. IPWD (Niu et al., 2022) also focuses on the gap between $D_t$ and $D_s$. However, different from OOD-KD, they mainly solve the imbalance in teachers' knowledge Some studies discuss the domain shift problem in cross-time object detection (Li et al., 2022; Yang et al., 2022), but grant access to $D_t$, which is impractical in real-world scenarios. These studies try to figure out the problems in the context of knowledge distillation. However, they either discuss a preliminary version of the problem or lack rigor in their analysis. In summary, it is crucial to recognize that there is a growing demand for solutions to OOD-KD, while the research in this area is still in its early stage.

## 3 PROBLEM FORMULATION

To illustrate the concept of Out-of-domain Knowledge Distillation, we focus on its application in image classification. In this work, the term "domain" refers to a set of input-label pairs denoted as $D = \{(x_i, y_i)\}_{i=1}^N$. Here, the input $x_i \in X \subset \mathbb{R}^{C \times H \times W}$ represents an image with $H \times W$ dimensions and $C$ channels, while the corresponding label is denoted as $y_i \in Y \subset \{0, 1, \cdots, K - 1\} := [K]$, where $K$ represents the number of classes. Vanilla KD methods guide the student model $S(\cdot; \theta_s)$ to imitate the teacher model $T(\cdot; \theta_t)$ and learn from the ground truth label, formatted as:

$$\hat{\theta}_s = \arg\min_{\theta_s} \mathbb{E}_{(x,y) \sim P_s} \left[ D_{\mathrm{KL}}(T(x; \theta_t) \parallel S(x; \theta_s)) + \mathrm{CE}(S(x; \theta_s), y) \right] \tag{1}$$

where CE refers Cross Entropy, $P_s$ is the joint distribution in $D_s$. In the context of OOD-KD, the teacher domain $D_t$ and the student domain $D_s$ differ in terms of the joint distribution $P(X, Y)$. For instance, in $D_t$, the majority of the images labeled as "cow" might depict cows on grassy landscapes. On the other hand, the ones in the student domain $D_s$ could show cows on beaches or other locations. Unavoidably, $T$ not only learns the class concept but also utilizes some spurious correlations (e.g., associating the background "grass" with the cow) to enhance its training performance.

However, as the occurrence of spurious correlations cannot be guaranteed in the target application, blindly mimicking the behavior of $T$ is unwise. Hence, the key challenge lies in leveraging the

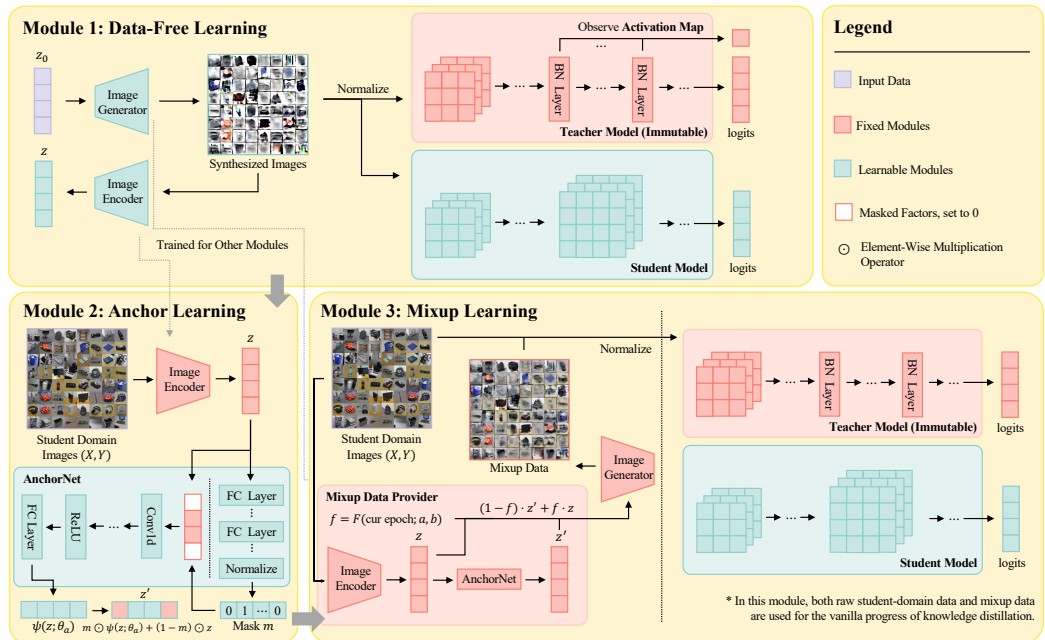

Figure 2: Overview of our proposed method, consisting of three major modules.

teacher's knowledge effectively, accounting for the domain shift between $D_s$ and $D_t$ (Zhang et al., 2022; 2023a; 2024a; Bai et al., 2024). Vanilla methods bridge this domain shift with the assistance of $T$'s training data. However, in OOD-KD, due to various reasons (privacy concerns, patent protection, computational resources, etc.), quite a number of models are released without granting access to their training data and even some of the models are hard to adapt. This situation further amplifies the difficulty of the problem. Hence, we present the definition of OOD-KD herein.

**Problem Definition:**  Given an **immutable** teacher model $T$ with its parameter $\theta_t$ and labeled student-domain data $D_s = \{(x_i, y_i)\}_{i=1}^{N_s}$ whose joint distribution $P(X, Y)$ **differs** from that of $D_t$ but the label space is the same ($Y_t = Y_s$), the objective of OOD-KD is to train a student model $S(\cdot; \theta_s)$ only with access to $D_s$ and $T$, leaving the teacher model $T$ unchanged in the overall process.

## 4   METHOLOGY

To address OOD-KD, we propose a simple but effective method AuG-KD. Generally, AuG-KD is composed of three modules: Data-Free Learning Module, Anchor Learning Module, and Miuxp Learning Module, as is vividly shown in Figure 2. In a certain module, the green blocks inside are trained with the help of fixed red blocks. The overall algorithm utilizes these 3 modules sequentially. For space issues, we leave the pseudo-code of our overall method in Appendix A. In the following sections, we provide detailed descriptions of each module.

### 4.1   MODULE 1: DATA-FREE LEARNING

To leverage $T$'s knowledge without access to its training data, DFKD methods are indispensable. This module follows the vanilla DFKD methods, training a Generator $G(\cdot; \theta_g) : Z \mapsto X$ from a normally-distributed latent variable $z_0 \sim \mathcal{N}(0, 1)$ under the instructions of the teacher model $T$. **The generated image is denoted as $x = G(z_0; \theta_g)$, while the normalized version of it is $\tilde{x} = N(x)$.** $y = \arg\max T(\tilde{x}; \theta_t)$ means the labels predicted by $T$. The dimension of $Z$ is denoted as $N_z$. H refers to the Information Entropy. AM stands for Activation Map, which observes the mean and variance of the outputs from BatchNorm2d layers.

$$L_{\text{KL}}(z_0) = D_{\text{KL}}(S(\tilde{x}; \theta_s) \parallel T(\tilde{x}; \theta_t)) \tag{2}$$

$$L_{\text{CE}}(z_0) = \text{CE}\big(T(\tilde{x}; \theta_t), y\big) \tag{3}$$

$$L_{\text{generator}} = \mathbb{E}_{z_0 \sim \mathcal{N}(0,1)} \Big[ -L_{\text{KL}} + L_{\text{CE}} + \alpha_g \cdot H(T(\tilde{x}; \theta_t)) + \text{AM}(T(\tilde{x}; \theta_t)) \Big] \tag{4}$$

$$\hat{\theta}_g = \arg\min_{\theta_g} L_{\text{generator}} \tag{5}$$

Meanwhile, an additional encoder $E(\cdot; \theta_e) : X, Y \mapsto Z$ is trained, keeping $\theta_g$ fixed. It absorbs the generated image $x = G(z_0; \theta_g)$ and the label $y = \arg\max T(\tilde{x}; \theta_t)$ as input and outputs the related latent variable $z = E(x, y; \theta_e)$ with Eq. 6, where MSE represents the mean squared error.

$$\hat{\theta}_e = \arg\min_{\theta_e} L_{\text{encoder}} = \arg\min_{\theta_e} \mathbb{E}_{z_0 \sim \mathcal{N}(0,1)} \Big[ \text{MSE}(z_0, z) + \alpha_e \cdot D_{\text{KL}}(z \parallel z_0) \Big] \tag{6}$$

When training the encoder, the student model $S$ is trained simultaneously with Eq. 7.

$$\hat{\theta}_s = \arg\min_{\theta_s} L_{\text{student}} = \arg\min_{\theta_s} \mathbb{E}_{z_0 \sim \mathcal{N}(0,1)}[L_{\text{KL}}] \tag{7}$$

## 4.2 MODULE 2: ANCHOR LEARNING

Anchor Learning Module trains an AnchorNet $(m, \psi; \theta_a)$ to map student-domain data to the teacher domain. It consists of a class-specific mask $m(\cdot; \theta_a) : Y \mapsto \{0, 1\}^{N_z}$ and a mapping function $\psi(\cdot; \theta_a) : Z \mapsto Z$, which are trained concurrently in this module. $m$ and $\psi$ are integrated into a lightweight neural network AnchorNet as shown in Figure 2. **Detailed implementations of them are provided in Appendix A.** This idea draws inspiration from invariant learning (Creager et al., 2021; Kuang et al., 2018), which is proposed especially for the problem of domain shift. IRM (Arjovsky et al., 2019) assumes the partial invariance either in input space or latent space, implying the presence of some invariant factors across domains despite the domain shift. In this work, we assume that **a portion of the latent variable $z$ exhibits such invariance**:

**Assumption 1** Given any image pair $((x_1, y_1), (x_2, y_2))$ that is **identical except for the domain-specific information**, there exists a **class-specific binary mask operator** $m(\cdot; \theta_a) : Y \mapsto \{0, 1\}^{N_z}$ that satisfies the partial invariance properties in the latent space under the Encoder $E(\cdot; \theta_e) : X, Y \mapsto Z$, as shown in Eq. 8. The mask masks certain dimensions in the latent space to zero if the corresponding component in it is set to 0 or preserves them if set to 1.

$$(\mathbf{1} - m(y_1; \theta_a)) \odot E(x_1, y_1; \theta_e) \equiv (\mathbf{1} - m(y_2; \theta_a)) \odot E(x_2, y_2; \theta_e) \tag{8}$$

$\odot$ in Eq 8 is the element-wise multiplication operator. Assumption 1 sheds light on the method of effectively transferring $T$'s knowledge: **just to change the domain-specific information**. With it, we can obtain the invariant part in the latent space of an arbitrary data sample. If we change the variant part, we can change the domain-specific information and thus can change the domain of the data sample to $D_t$. As a result, $T$ can provide more useful information for distillation. To direct $\psi$ to change the domain-specific information and map the samples to $D_t$, we introduce the uncertainty metric $U(x; T)$ which draws inspiration from Energy Score (Liu et al., 2020), formulated as:

$$U(x; T) = -t \cdot \log \sum_i^K \exp \frac{T_i(x)}{t} \tag{9}$$

where $t$ is the temperature and $T_i(x)$ denotes the $i^{\text{th}}$ logits of image $x$ output by the teacher model $T$. $U(x; T)$ measures $T$'s uncertainty of an arbitrary image $x$. The lower the value of $U(x; T)$ is, the more confident $T$ is in its prediction.

To preserve more semantic information during the mapping, we include the cross-entropy between $T$'s prediction on the mapped image and the ground truth label in the loss function of AnchorNet, as shown in Eq. 10-11, where $x' = G(z'; \theta_g)$ and $z' = m(y; \theta_a) \odot \psi(z; \theta_a) + (1 - m(y; \theta_a)) \odot z$ represent the resultant images and latent variables after mapping individually. We denote $x'$ as "anchor". These anchors are in the teacher domain $D_t$. $T$ is hence more confident about its prediction on them and can thus provide more useful information for distillation.

For simplicity, the portion of invariant dimensions in $z$ is preset by $\alpha_a$. $L_{\text{inv}}$ regulates it based on the absolute error between the $l_1$-norm and the desired number of ones in the mask $m$.

$$L_{\text{inv}}(y) = |(1 - \alpha_a) \cdot N_z - \|m(y; \theta_a)\|_1| \tag{10}$$

$$\hat{\theta}_a = \arg\min_{\theta_a} L_{\text{anchor}} = \arg\min_{\theta_a} \mathbb{E}_{(x,y) \sim P_s} \Big[ U(x'; T) + L_{\text{inv}}(y) + \beta_a \cdot \text{CE}(T(x'; \theta_t), y) \Big] \tag{11}$$



| f = 0 | f = 0.25 | f = 0.5 | f = 0.75 | f = 1 |

Figure 3: Different mixup samples generated in Module 3 for DSLR in Office-31, controlled by the stage factor $f \in [0, 1]$. The value of $f$ determines the proximity of the samples to $D_t$ and $D_s$. A smaller value of $f$ indicates that the samples are closer to the teacher domain $D_t$, while a larger value of $f$ indicates that the samples are closer to the student domain $D_s$.

## 4.3 MODULE 3: MIXUP LEARNING

This module is a process of knowledge distillation using $D_s$ and mixup data provided by AnchorNet. To be specific, for an arbitrary image $x$ in $D_s$, Encoder $E(\cdot; \theta_e)$ encodes it to $z$ and AnchorNet $(m, \psi; \theta_a)$ maps it to $z'$. Mixup Learning utilizes the mapping from $z'$ to $z$ to generate a series of images that evolves during the training process. The evolution is governed by a stage factor $f$, which is given by a monotonically non-decreasing scheduler function $F(\cdot; a, b) : \mathbb{N} \mapsto [0, 1]$. Parameter $a$ controls the rate of the change of mixup images, while $b$ determines their starting point. These parameters adhere to the property $F(a \cdot \sharp\text{Epoch}; a, b) = 1$ and $F(0; a, b) = b$, where $\sharp\text{Epoch}$ represents the total number of training epochs. The mixup samples are formulated as shown in Eq. 12. Figure 3 vividly illustrates the mixup samples provided.

$$x_{\mathrm{m}} = (1 - f) \cdot G((1 - f) \cdot z' + f \cdot z; \theta_g) + f \cdot x \qquad (12)$$

As the latent variable evolves from $z'$ to $z$, the mixup samples evolve from $D_t$ to $D_s$. Consequently, at the beginning of training, the teacher model $T$ exhibits more certainty regarding the samples and can provide more valuable knowledge. As training progresses, $T$ becomes less certain about its predictions, which thus encourages the student model to learn more from the student-domain data.

Table 1: Test accuracies (%) of distilling ResNet34 to MobileNet-V3-Small on Office-31. The row "Settings" implies the arrangement of domains. For instance, "Amazon, Webcam→DSLR" indicates that $T$ is trained on Amazon and Webcam, $S$ is to be adapted on DSLR. The first row of Teacher is $T$'s performance on $D_t$, while the second row is that on $D_s$. The "+" mask signifies that these methods are fine-tuned on the student domain after applying the respective methods.

| | Office-31: resnet34 → mobilenet_v3_small | | | | | | | | |
|---|---|---|---|---|---|---|---|---|---|
| **Settings** | **Amazon, Webcam→DSLR** | | | **Amazon, DSLR→Webcam** | | | **DSLR, Webcam→Amazon** | | |
| | **Acc** | **Acc@3** | **Acc@5** | **Acc** | **Acc@3** | **Acc@5** | **Acc** | **Acc@3** | **Acc@5** |
| **Teacher** | 92.2 | 96.1 | 97.0 | 93.1 | 96.2 | 97.3 | 97.7 | 99.6 | 99.8 |
| | 67.1 | 82.6 | 88.0 | 60.0 | 77.5 | 82.5 | 15.2 | 26.1 | 36.0 |
| **DFQ+** | 80.4±5.7 | 93.3±4.1 | 96.4±2.1 | 86.5±5.7 | **97.5±2.0** | 99.0±1.0 | 46.6±4.5 | 67.6±2.4 | 76.5±2.9 |
| **CMI+** | 67.1±3.5 | 86.6±4.3 | 92.9±3.0 | 70.0±5.3 | 88.0±5.1 | 94.3±2.1 | 35.9±2.3 | 56.1±5.1 | 65.0±5.6 |
| **DeepInv+** | 65.9±6.3 | 84.7±4.9 | 90.6±3.8 | 70.0±5.4 | 91.5±0.5 | 94.8±1.6 | 36.5±4.4 | 56.1±5.1 | 66.3±3.3 |
| **w/o KD** | 63.5±7.9 | 84.7±4.5 | 90.2±3.7 | 82.7±5.4 | 96.0±1.9 | 98.3±0.7 | 52.9±3.4 | 72.5±3.6 | **79.9±2.2** |
| **ZSKT+** | 33.3±5.9 | 55.3±11.8 | 65.9±11.5 | 33.0±8.1 | 55.3±14.3 | 66.8±16.2 | 23.7±5.3 | 42.7±7.1 | 53.7±5.9 |
| **PRE-DFKD+** | 68.3±19.5 | 87.8±14.3 | 91.8±13.3 | 66.5±20.9 | 82.0±17.3 | 88.9±12.9 | 28.4±13.3 | 46.4±19.0 | 55.9±20.8 |
| **Ours** | **84.3±3.1** | **94.9±2.6** | **97.6±0.8** | **87.8±7.6** | 96.3±1.8 | **99.5±0.7** | **58.8±3.7** | **73.7±2.1** | 79.7±1.5 |

## 5 EXPERIMENTS

### 5.1 EXPERIMENTAL SETTINGS

The proposed method is evaluated on 3 datasets Office-31 (Saenko et al., 2010), Office-Home (Venkateswara et al., 2017), and VisDA-2017 (Peng et al., 2017). These datasets consist of multiple domains and are hence appropriate to our study.

**Office-31** This dataset contains 31 object categories in three domains: Amazon, DSLR, and Webcam with 2817, 498, and 795 images respectively, different in background, viewpoint, color, etc.

**Office-Home** Office-Home is a 65-class dataset with 4 domains: Art, Clipart, Product, and Real-World. Office-Home comprises 15500 images, with 70 images per class on average.

Table 2: Test accuracies (%) of distilling ResNet34 to MobileNet-V3-Small on dataset Office-Home. A, C, P, and R refer to Art, Clipart, Product, and Real-World respectively.

| | Office-Home: resnet34 → mobilenet_v3_small | | | | | | | | | | | |
|---|---|---|---|---|---|---|---|---|---|---|---|---|
| Settings | ACP→R | | | ACR→P | | | APR→C | | | CPR→A | | |
| | Acc | Acc@3 | Acc@5 | Acc | Acc@3 | Acc@5 | Acc | Acc@3 | Acc@5 | Acc | Acc@3 | Acc@5 |
| Teacher | 89.4 | 93.2 | 94.5 | 88.6 | 92.5 | 93.8 | 66.7 | 75.8 | 79.6 | 90.9 | 94.4 | 95.4 |
| | 30.3 | 46.7 | 55.2 | 35.8 | 53.4 | 60.8 | 24.7 | 40.6 | 48.7 | 19.6 | 31.2 | 39.1 |
| DFQ+ | 33.3±1.3 | 51.7±1.4 | 60.7±1.7 | 60.0±3.8 | 75.8±2.9 | 81.8±2.6 | 50.6±2.8 | 67.7±2.8 | 75.2±1.6 | 21.0±3.4 | 31.8±3.5 | 40.3±2.5 |
| CMI+ | 16.4±1.2 | 29.0±0.4 | 37.0±0.7 | 48.8±1.5 | 63.9±1.4 | 70.3±1.6 | 35.3±1.9 | 51.2±2.0 | 58.4±1.7 | 13.4±3.0 | 21.4±2.7 | 27.5±2.8 |
| DeepInv+ | 15.4±1.7 | 28.6±1.8 | 36.4±2.1 | 47.8±2.1 | 62.9±2.2 | 70.7±2.2 | 36.9±2.5 | 52.5±3.4 | 60.5±2.9 | 13.0±2.1 | 22.3±2.4 | 27.5±2.1 |
| w/o KD | 32.5±3.8 | 48.4±3.8 | 57.6±3.3 | 59.9±2.0 | 77.2±1.4 | 82.6±0.8 | 49.9±1.4 | 67.0±1.6 | 73.6±1.1 | 16.8±2.1 | 28.9±1.5 | 36.4±2.3 |
| ZSKT+ | 15.5±3.3 | 29.7±4.4 | 38.5±4.5 | 11.9±5.6 | 23.5±9.5 | 32.0±10.9 | 7.8±2.9 | 19.5±5.3 | 27.5±6.7 | 7.9±3.1 | 17.6±3.6 | 26.7±3.0 |
| PRE-DFKD+ | 22.3±3.7 | 36.9±5.0 | 44.9±5.2 | 34.4±9.5 | 52.4±11.2 | 60.5±11.0 | 38.4±7.9 | 57.9±11.2 | 65.4±11.1 | 9.0±2.7 | 20.4±3.7 | 27.5±5.0 |
| Ours | 35.2±2.5 | 53.4±2.0 | 62.8±1.8 | 65.3±1.6 | 79.3±1.4 | 84.1±2.0 | 53.4±3.0 | 70.3±1.4 | 76.6±1.4 | 21.2±4.7 | 33.4±3.8 | 41.7±4.6 |

Table 3: Test accuracies (%) of distilling ResNet34 to MobileNet-V3-Small on dataset VisDA-2017. Methods are adapted on the validation domain and tested on the test domain. The first column of Teacher is the results in the train domain, while the second row is those in the validation domain.

| | VisDA-2017 (train→validation): resnet34 → mobilenet_v3_small | | | | | | | |
|---|---|---|---|---|---|---|---|---|
| Settings | Teacher | | DFQ+ | CMI+ | DeepInv+ | w/o KD | ZSKT+ | PRE-DFKD+ | Ours |
| Acc | 100.0 | 12.1 | 53.4±1.0 | 49.5±1.3 | 47.6±0.9 | 50.7±0.9 | 48.4±3.5 | 54.9±1.0 | **55.5±0.3** |
| Acc@3 | 100.0 | 34.5 | 80.2±0.6 | 77.2±1.1 | 75.5±0.7 | 78.7±0.7 | 77.5±2.6 | 81.4±1.0 | **82.1±0.2** |
| Acc@5 | 100.0 | 54.7 | 89.0±0.4 | 88.1±0.7 | 87.3±0.6 | 89.3±0.4 | 88.9±1.2 | 90.6±0.6 | **91.3±0.1** |

**VisDA-2017** VisDA-2017 is a 12-class dataset with over 280000 images divided into 3 domains: train, validation, and test. The training images are simulated images from 3D objects, while the validation images are real images collected from MSCOCO (Lin et al., 2014).

Main experiments adopt ResNet34 (He et al., 2016) as the teacher model and MobileNet-V3-Small (Howard et al., 2019) as the student model. Usually, teacher models are trained with more data samples (maybe from multiple sources) than student models. **To better align with real-world scenarios, all domains are utilized for training the teacher model $T$, except for one domain that is reserved specifically for adapting the student model $S$.** Since Office-31 and Office-Home do not have official train-test splits released, for evaluation purposes, the student domain $D_s$ of these two datasets is divided into training, validation, and testing sets using a seed, with proportions set at 8:1:1 respectively. As to VisDA-2017, we split the validation domain into 80% training and 20% validation and directly use the test domain for test. The performance of our methods is compared with baselines using top 1, 3, and 5 accuracy metrics.

Given that OOD-KD is a relatively novel problem, there are no readily available baselines. Instead, we adopt state-of-the-art DFKD methods, including DFQ (Choi et al., 2020), CMI (Fang et al., 2021b), DeepInv (Yin et al., 2020), ZSKT (Micaelli & Storkey, 2019), and PRE-DFKD (Binici et al., 2022), and fine-tune them on the student domain. One more baseline "w/o KD" is to train the student model $S$ without the assistance of $T$, starting with weights pre-trained on ImageNet (Deng et al., 2009). To ensure stability, each experiment is conducted five times using different seeds, and the results are reported as mean ± standard variance.

Due to limited space, we leave **hyperparameter settings, full ablation results, and combination with other baselines (like Domain Adaptation methods)** to Appendix B and C.

## 5.2 Results and Observations

Our main results are summarized in Table 1, 2, and 3. Extensive experiments solidly substantiate the stability and superiority of our methods. In this section, we will discuss the details of our results.

**Larger domain shift incurs larger performance degradation.** It is evident that all the teacher models experience significant performance degradation when subjected to domain shift. The extent of degradation is directly proportional to the dissimilarity between the student domain $D_s$ and the teacher domain $D_t$. For instance, in Office-Home, Art is the most distinctive domain, it is significantly different from other domains. As a result, in the CPR→A setting, the performance of the teacher model $T$ exhibits the largest decline, with an approximate 70% drop absolutely. The same

phenomenon can be observed in VisDA-2017, where the train domain is 3D simulated images, but the others are real-world photographs. Moreover, the problem of performance degradation can be amplified by the imbalance amount of training data between $T$ and $S$. Usually, we assume that teacher models are trained with more data samples. When the assumption violates, like DW $\rightarrow$ A in Office-31, where the Amazon domain is larger than the sum of other domains, the issue of performance degradation becomes more prominent.

**DFKD methods are unstable but can be cured with more data samples.** It is worth noting that the standard variance of each method, in most settings, is slightly high. This observation can be attributed to both the inherent characteristics of DFKD methods and the limited amount of data for adaptation. As DFKD methods train a generator from scratch solely based on information provided by the teacher model, their stability is not fully guaranteed. Although **the remedy to it goes beyond our discussion**, it is worth noting that as the amount of data increases (Office-31 (5000) to VisDA-2017 (28000)), these methods exhibit improved stability (Office-31 (7.6) to VisDA-2017 (0.3)).

**AnchorNet DOES change the domain of data samples.** To make sure AnchorNet does enable $T$ to provide more useful information, we observe the mixup data samples in Mixup Learning Module under the setting Amazon, Webcam $\rightarrow$ DSLR (AW $\rightarrow$ D) in Office-31, as shown in Figure 3. These domains exhibit variations in background, viewpoint, noise, and color. Figure 1 gives a few examples in Amazon (the right up) and DSLR (the right bottom). Obviously, Amazon differs from other domains – the background of the samples in it is white. In AW $\rightarrow$ D, when $f = 0$ the images are closer to $D_t$, with white backgrounds (Amazon has more samples than Webcam). As $f$ goes larger, the mixup samples get closer to $D_s$, depicting more features of DSLR.

## 5.3 ABLATION STUDY

To further validate the effectiveness of our methods, we perform ablation experiments from three perspectives: Framework, Hyperparameter, and Setting. In line with our main experiments, each experiment is conducted five times with different seeds to ensure the reliability of the results. For simplicity, we focus on Amazon, Webcam $\rightarrow$ DSLR setting in Office-31.

(a) Ablation study on the framework of our method.

| Framework Ablation: Amazon, Webcam $\rightarrow$ DSLR | | | |
|---|---|---|---|
| **Method** | **Acc** | **Acc@3** | **Acc@5** |
| **M1** | 33.7±5.1 | 56.1±8.1 | 68.6±3.1 |
| **M1+M2+M3 (w/o Mixup)** | 80.1±5.7 46.4↑ ±0.6↑ | 92.3±4.1 36.2↑ ±4.0↓ | 96.4±2.1 27.8↑ ±1.0↓ |
| **M1+M3** | 83.5±4.1 3.4↓ ±1.6↓ | 94.5±2.9 3.2↑ ±1.2↓ | 96.1±2.0 0.3↓ ±0.1↓ |
| **M1+M2+M3** | 84.3±3.1 0.8↑ ±1.0↓ | 94.9±2.6 0.4↑ ±0.3↓ | 97.6±0.8 1.6↑ ±1.1↓ |

(b) Ablation study (Acc) on different $T \rightarrow S$ pairs.

| Setting Ablation: Amazon, Webcam $\rightarrow$ DSLR | | | | |
|---|---|---|---|---|
| **T $\rightarrow$ S pair** | r34$\rightarrow$ mb | r50 $\rightarrow$ r18 | r34 $\rightarrow$ sf | r34 $\rightarrow$ ef |
| **DFQ+** | 80.4±5.7 | 87.5±5.7 | 86.3±2.4 | 90.2±4.6 |
| **w/o KD** | 63.5±7.9 | 84.3±4.8 | 79.6±1.8 | 85.1±4.7 |
| **PRE-DFKD+** | 68.3±19.5 | 79.2±5.6 | 87.9±5.1 | 83.9±4.9 |
| **Ours** | 84.3±3.1 | 88.2±4.3 | 88.6±5.1 | 91.8±3.5 |

### 5.3.1 FRAMEWORK ABLATION

Here, we evaluate the effectiveness of our modules. Framework ablation studies traditionally involve masking parts of the proposed modules for experimental purposes. Yet, it is essential to recognize: 1. **Module 1 is fundamental to our method and is non-removable**; 2. Module 2 serves to support Module 3. **There is no need to test the results only with Module 1&2**. Hence, our investigation focuses on the outcomes absent Module 2, and absent both Module 2 and 3, denoted as M1+M3 and M1, respectively. Additionally, our analysis dives into Module 3, where **we omit the mixup samples to evaluate their critical role, denoted as M1+M2+M3 (w/o Mixup)**. It's worth noting that **there is no need to add one more setting w/o M2 & Mixup here** since it makes no difference to M1+M2+M3 (w/o Mixup). Consequently, we get three distinct ablation scenarios: M1, M1+M3, and M1+M2+M3 (w/o Mixup). To be specific, in M1, we directly choose $S$'s best checkpoint in Module 1 and test it on $D_s$. In M1+M2+M3 (w/o Mixup), the model trains solely on $D_s$. In M1+M3, we mask AnchorNet by equating its output $z'$ with its input $z$ and then proceed with the method.

The results are presented in Table 4a. The performance improvement between M1 and M1+M2+M3 (w/o Mixup) mainly stems from the supervision of $D_s$. As M1 is a simple DFKD, the striking performance gap underscores the urgent need for solutions to OOD-KD. The considerable enhancement

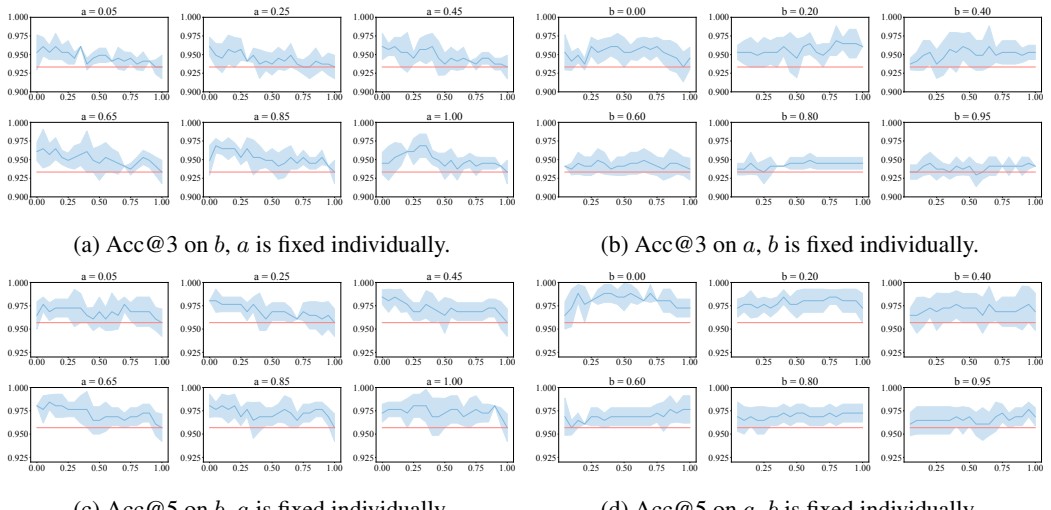

(a) Acc@3 on $b$, $a$ is fixed individually.

(b) Acc@3 on $a$, $b$ is fixed individually.

(c) Acc@5 on $b$, $a$ is fixed individually.

(d) Acc@5 on $a$, $b$ is fixed individually.

Figure 4: Grid study on hyperparameter $a$ and $b$ in Module 3. The red line is $b = 1.0$, meaning no mixup data. The blue line portrays the performance of various $a - b$ settings. The light blue area symbolizes the range encompassing mean $\pm$ std.

evidences the efficacy of Module 2 and 3 in remedying domain shifts. The rise in average accuracy coupled with reduced variance firmly attests to the significance of each component in our method.

### 5.3.2 SETTING ABLATION

In this study, we change the $T - S$ pair in our experiments. We additionally employ ResNet50 $\rightarrow$ ResNet18 (r50 $\rightarrow$ r18), ResNet34 $\rightarrow$ ShuffleNet-V2-X0-5 (r34 $\rightarrow$ sf) and ResNet34 $\rightarrow$ EfficientNet-B0 (r34 $\rightarrow$ ef) in this study. The ResNet50 $\rightarrow$ ResNet18 pair is a commonly used evaluation pair in traditional distillation methods, while ShuffleNet and EfficientNet are well-known lightweight neural networks suitable for edge devices. These pairs are compared with several effective baselines in our main experiments. The results of this study are displayed in Table 4b, which confirm the effectiveness of our methods across different teacher-student distillation pairs.

### 5.3.3 HYPERPARAMETER ABLATION

In this ablation study, our primary focus is on two hyperparameters, namely $a$ and $b$ in Module 3, which govern the speed and starting point of the mixup data samples. We perform a grid study on the values of $a$ and $b$ within their domain $[0, 1]$, with a step size of 0.05. Since $a = 0$ is useless but causes the division-by-zero problem, we set the minimum value of $a$ to step size 0.05.

Detailed results are depicted in Figure 4. Due to the limited space, we present only a portion of $a - b$ assignments here, with more results included in Appendix C. The red line in the figures represents the baseline, wherein no mixup data but only raw images are provided. Notably, the blue line consistently surpasses the red line over the majority of the range, testifying to the effectiveness of our method. Both Figure 4a and 4c demonstrate a slight decrease in performance as $b$ increases, suggesting that an excessively large assignment of $b$ is not preferred.

## 6 CONCLUSION

In this work, we dive into the problem of Out-of-Domain Knowledge Distillation to selectively transfer teachers' proper knowledge to students. Further, we propose a simple but effective method AuG-KD. It utilizes a data-driven anchor to align student-domain data with the teacher domain and leverages a generative method to progressively evolve the learning process from OOD knowledge distillation to domain-specific information learning. Extensive experiments validate the stability and superiority of our approach. However, it is worth emphasizing that the research on OOD-KD is still in its early stages and considered preliminary. Therefore, we encourage further attention and exploration in this emerging and practical field.

## ACKNOWLEDGMENTS

This work was supported by National Science and Technology Major Project (2022ZD0119100), the National Natural Science Foundation of China (62376243, 62037001, U20A20387), Scientific Research Fund of Zhejiang Provincial Education Department (Y202353679), and the StarryNight Science Fund of Zhejiang University Shanghai Institute for Advanced Study (SN-ZJU-SIAS-0010).

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

## A  METHOD DETAILS

Our proposed method consists of three main modules. Data-Free Learning Module serves as the cornerstone of the entire approach. It trains a teacher domain $D_t$'s generator $G(\cdot; \theta_g) : Z \mapsto X$ and encoder $E(\cdot; \theta_e) : X, Y \mapsto Z$ and warmups the student model $S$ in advance. In Anchor Learning Module, a mapping is established within the latent space $Z$ that aligns the distribution of $D_s$ to that of $D_t$, taking into account the uncertainty metric provided by $T$. Finally, in Mixup Learning Module, the mapping obtained in Anchor Learning Module is employed to generate synthetic data of $D_t$ and mixuped with $D_s$. The pseudo-code of our proposed method is displayed in Algorithm 1.

---

**Algorithm 1:** Pseudo-code of our proposed method

---

**Input:** Student domain data $D_s$, Batch size $b$, Latent size $N_z$
**Output:** Optimized $S(\cdot; \theta_s)$
```
/* Data-Free Learning Module, trains G(·;θ_g), E(·;θ_e), S(·;θ_s)     */
```
Sample $z_\text{val}$ from normal distribution, with size $(10b, N_z)$;
**for** $i \leftarrow 1$ *to ♯epoch* **do**
    Sample $z_0$ from normal distribution with size $(b, N_z)$;
    Compute $L_\text{generator}$ and update $\theta_g$;
    **for** $\_ \leftarrow 1$ *to 5* **do**
        Compute $L_\text{encoder}, L_\text{student}$ and update $\theta_e, \theta_t$;
    **end**
    Evaluate $G(\cdot; \theta_g), E(\cdot; \theta_e), S(\cdot; \theta_s)$ with $z_\text{val}$, save the best parameter;
**end**
```
/* Anchor Learning Module, trains (m,ψ;θ_a)                           */
```
**for** $i \leftarrow 1$ *to ♯epoch* **do**
    **for** $(x, y) \in D_s$*'s training set* **do**
        $z \leftarrow E(x, y; \theta_e)$;
        $z' \leftarrow m(y; \theta_a) \odot \psi(z; \theta_a) + (1 - m(y; \theta_a)) \odot z$;
        $x' \leftarrow G(z'; \theta_g)$;
        Compute $L_\text{anchor}$ and update $\theta_g$
    **end**
    Evaluate $(m, \psi; \theta_a)$ with $D_s$'s validation set, save the best parameter;
**end**
```
/* Mixup Learning Module, trains S(·;θ_s)                             */
```
**for** $i \leftarrow 1$ *to ♯epoch* **do**
    **for** $(x, y) \in D_s$*'s training set* **do**
        $f \leftarrow F(i - 1; a, b)$;
        $x_\text{m} \leftarrow (1 - f) \cdot G(f \cdot z' + (1 - f) \cdot z; \theta_g) + f \cdot x$;
        $(x, y) \leftarrow (x || x_m, y || y)$`/* Concatenate two batches          */`
        Compute $L_\text{student}$ with newly get $(x, y)$ and update $\theta_s$
    **end**
    Evaluate $S(\cdot; \theta_s)$ with $D_s$'s validation set, save the best parameter;
**end**

---

In Anchor Learning Module, we integrate the mask operator $m$ and the mapping function $\psi$ into a lightweight neural network AnchorNet. Concretely, the network is implemented with Pytorch as shown in Code A. In a forward pass, we first embed the class label to get the class-specific mask and then map the latent variable back to $D_t$. To retrain domain-invariant information during mapping, we combine the mapped latent variable with the original one with the help of the class-specific mask.

```python
class AnchorNet(nn.Module):
    """AnchorNet

    Args:
        latent_size (int): Latent dimensionality
        num_classes (int): Number of classes
```

```
    The AnchorNet module takes an input tensor and a label tensor as input.

    It embeds the class labels, generates a mask based on the embedding,
    masks the input, and passes it through a CNN module.

    The CNN module consists of 1D convolutional and linear layers.

    The weights are initialized from a uniform distribution in __init__.

    The forward pass:
      1. Embeds class labels
      2. Generates mask from label embedding
      3. Masks input tensor
      4. Passes masked input through CNN module
      5. Returns masked output and mask tensor

    """
    def __init__(self, latent_size: int, num_classes: int):
        super().__init__()

        self.num_classes = num_classes
        self.embed_class = nn.Linear(num_classes, latent_size)

        self.mask = nn.Sequential(
            nn.Linear(latent_size, latent_size),
            nn.Linear(latent_size, latent_size),
            nn.Linear(latent_size, latent_size),
            nn.BatchNorm1d(latent_size),
            nn.Sigmoid(),
            Lambda(lambda x: x - 0.5),
            nn.Softsign(),
            nn.ReLU()
        )

        self.module = nn.Sequential(
            View(1, -1),
            nn.Conv1d(1, 4, 3, 1, 1),
            nn.BatchNorm1d(4),
            nn.LeakyReLU(),
            nn.Conv1d(4, 8, 3, 1, 1),
            nn.BatchNorm1d(8),
            nn.LeakyReLU(),
            nn.Conv1d(8, 4, 3, 1, 1),
            nn.BatchNorm1d(4),
            nn.LeakyReLU(),
            View(-1),
            nn.Linear(4 * latent_size, latent_size),
        )

        ... (initializations)

    def forward(self, inputs: Tensor, **kwargs) -> Tuple[Tensor, Tensor]:
        y = self.embed_class(one_hot(kwargs['labels'], self.num_classes))
        mask = self.mask(y)

        masked_inputs = inputs * mask
        z = self.module(masked_inputs)
        return masked_inputs * z + (1 - masked_inputs) * inputs, mask
```

In Mixup Learning Module, the generation of mixup samples is controlled by a monotonically non-decreasing scheduler function $F(\cdot; a, b) : \mathbb{N} \mapsto [0, 1]$, which is parameterized by $a$ and $b$. Parameter $a$ controls the rate of the change of mixup images, while $b$ determines their starting point. These parameters adhere to the property $F(a \cdot \sharp\text{Epoch}; a, b) = 1$ and $F(0; a, b) = b$. The idea of scheduler function draws inspiration from Curriculum Learning (Wang et al., 2019). All of our experiments directly adopt the simplest linear scheduler function:

$$F(x; a, b) = \frac{1-b}{a \cdot \sharp\text{Epoch}} \cdot \min(\max(x, 0), a \cdot \sharp\text{Epoch}) + \text{b}$$

## B  EXPERIMENT DETAILS

Each experiment is conducted using a single NVIDIA GeForce RTX 3090 and takes approximately 1 day to complete.

### B.1  HYPERPARAMETERS AND TRAINING SCHEDULES

We summarize the hyperparameters and training schedules of AuG-KD on the three datasets in Table 5.

Table 5: Hyperparameters and training schedules of AuG-KD.

| Dataset | Parameters | Setting |
|---|---|---|
| | GPU | NVIDIA GeForce RTX 3090 |
| | Optimizer | Adam |
| | Learning Rate (except Encoder) | 1e-3 |
| | Learning Rate (Encoder)GPI | 1e-4 |
| Offie-31 | Batch size | 2048 |
| Office-Home | $N_z$ | 256 |
| VisDA-2017 | Image Resolution | 32×32 |
| | seed | {2021,2022,$\cdots$,2025} |
| | $\alpha_g$ | 20 |
| | $\alpha_e$ | 0.00025 |
| | $\alpha_a$ | 0.25 |
| | $\beta_a$ | 0.1 |

Notably, the temperature of the KL-divergence in Module 3 is set to 10. As to baselines, we adopt their own hyperparameter settings. During the fine-tuning stage of each baseline, the standard setting involves 200 epochs, with a learning rate of 1e-3 and weight decay of 1e-4. Slight adjustments for optimal results are granted.

Module 3 is determined by two significant hyperparameters $a$ and $b$, which control mixup data's evolution speed and starting point. In the section of the ablation study, we have demonstrated that most $a - b$ settings are effective. For reproducibility, we provide detailed $a - b$ assignments in our main experiments in Table 6.

Table 6: Detailed $a - b$ assignments in our main experiments. The column Setting gives the domains $T$ and $S$ use. In Office-31, A means Amazon, W means Webcam, and D means DSLR individually. In Office-Home, A means Art, C means Clipart, P means Product, and R means Real-World individually.

| $a - b$ Setting in Main Experiments | | | |
|---|---|---|---|
| **Dataset** | **Setting** | **a** | **b** |
| **Office-31** | AW→D | 0.6 | 0.2 |
| | AD→W | 0.4 | 0.6 |
| | DW→A | 0.8 | 0.2 |
| **Office-Home** | ACP→R | 0.4 | 0.6 |
| | ACR→P | 0.8 | 0.2 |
| | APR→C | 0.8 | 0.2 |
| | CPR→A | 0.8 | 0.2 |
| **VisDA-2017** | train→val | 0.8 | 0.2 |

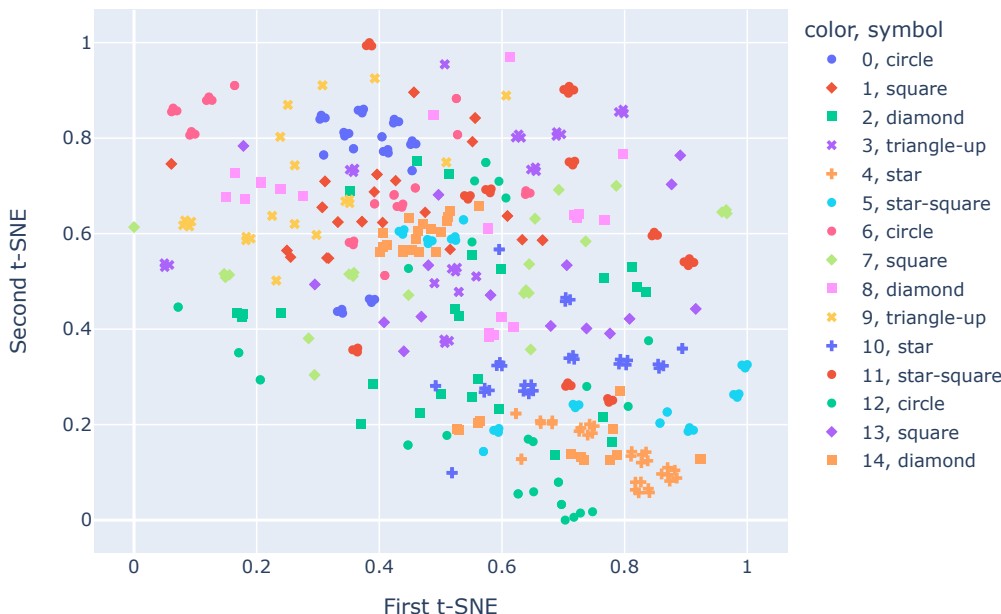

Figure 5: Visualization Results on $z$

## C  ADDITIONAL RESULTS

### C.1  VISUALIZATION ON MASK PROVIDED BY ANCHORNET

In Module 2, AnchorNet integrates the mask operator $m$ and the mapping function $\psi$ into a lightweight neural network. The mask is class-specific and **plays a crucial role in retaining domain-invariant knowledge in the latent space**. To vividly demonstrate the effectiveness of the mask, we conduct t-SNE (van der Maaten & Hinton, 2008) on the latent variables and the masked version of them. To be specific, we use AnchorNet and Encoder trained under the setting AW→D in Office-31 and select 32 images for each class (31 classes in total). For each image $(x, y)$, we encode it to get the latent variable $z = E(x; \theta_e)$, and obtain the class-specific mask $m(y; \theta_a)$. Next, we obtain the masked latent variable $z' = (1 - m(y; \theta_a)) \odot z$. The t-SNE results on $z$ and $z'$ are displayed in Figure 5 and 6 Each displays a distribution of data points plotted against two t-SNE components. The points are colored and shaped differently to represent different classes within the latent space.

In Figure 5, the distribution is quite mixed, with no distinct clusters or separation between the different classes. In contrast, in Figure 6, after applying mask operation on the latent variables, there appears to be a more distinct separation between different classes. Clusters of the same shapes and colors are more evident, indicating that the mask operation has enhanced the class-specific knowledge within the latent space.

### C.2  FULL ABLATION STUDY RESULTS

In previous sections, we thoroughly examined the impact of various assignments of $a$ and $b$ on the overall performance. For the sake of limited space, we only demonstrate part of the results previously. Full results are provided in Figure 7-12. The red line in the figures represents the baseline, wherein no mixup data but only raw images are provided. These results are in alignment with the observations before. Notably, the blue line consistently surpasses the red line over the

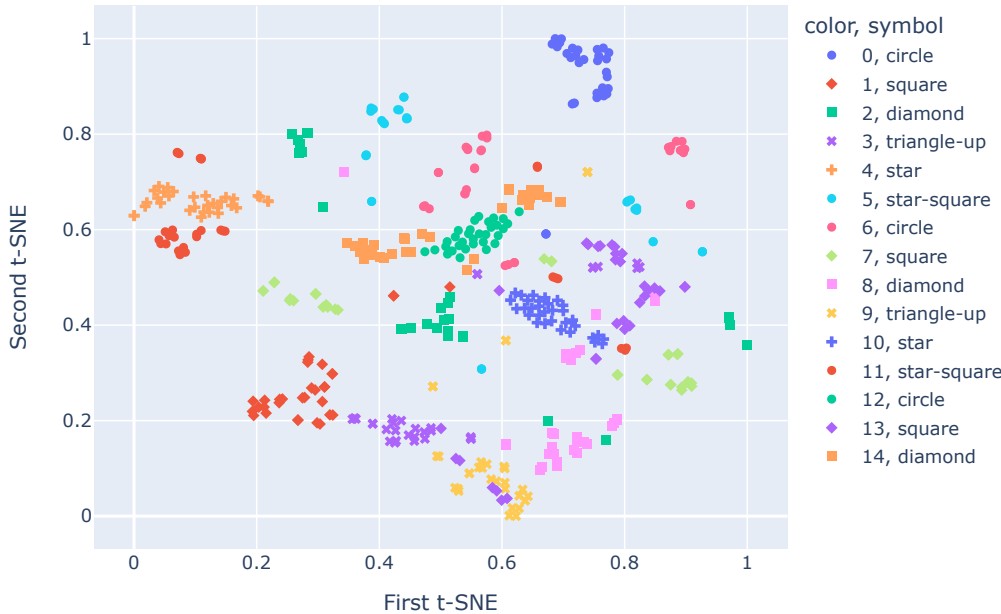

Figure 6: Visualization Results on $z'$

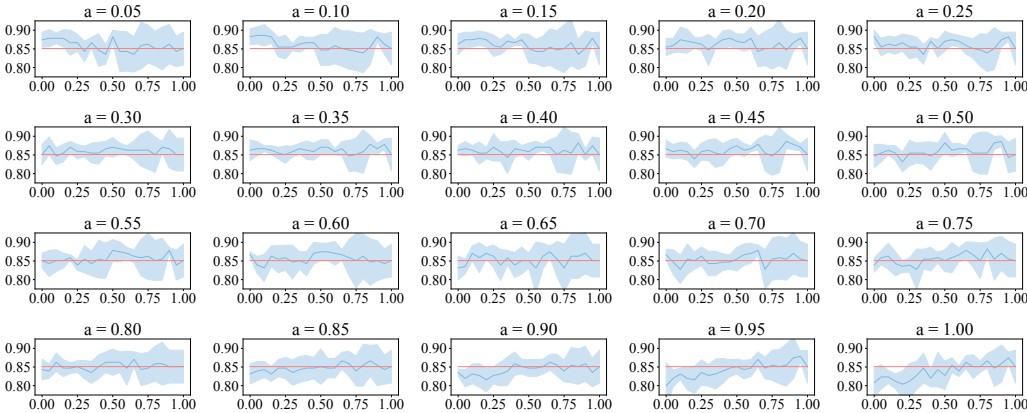

Figure 7: Grid study on hyperparameter $a$ and $b$ in Module 3. The red line is $b = 1.0$, meaning no mixup data. The blue line portrays the performance of various $a - b$ settings. The light blue area symbolizes the range encompassing mean $\pm$ std. This figure is the ablation results of Acc@1 on $b$ with $a$ fixed individually.

majority of the range. Most $a - b$ assignments provide effective mixup samples that better transfer the knowledge of the teacher model.

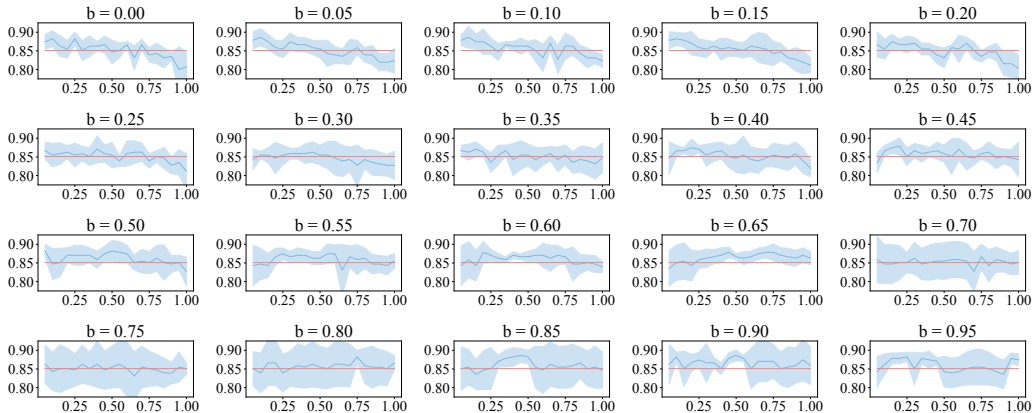

Figure 8: Acc@1 on $b$, $a$ is fixed individually.

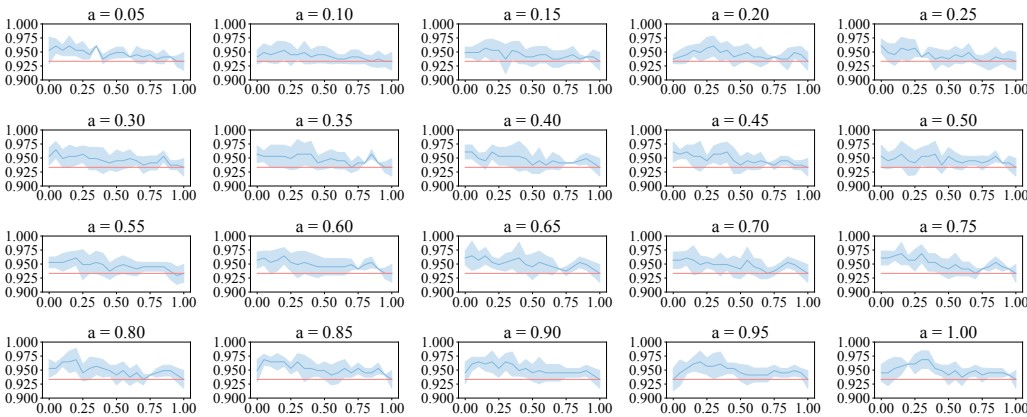

Figure 9: Acc@3 on $b$, $a$ is fixed individually.

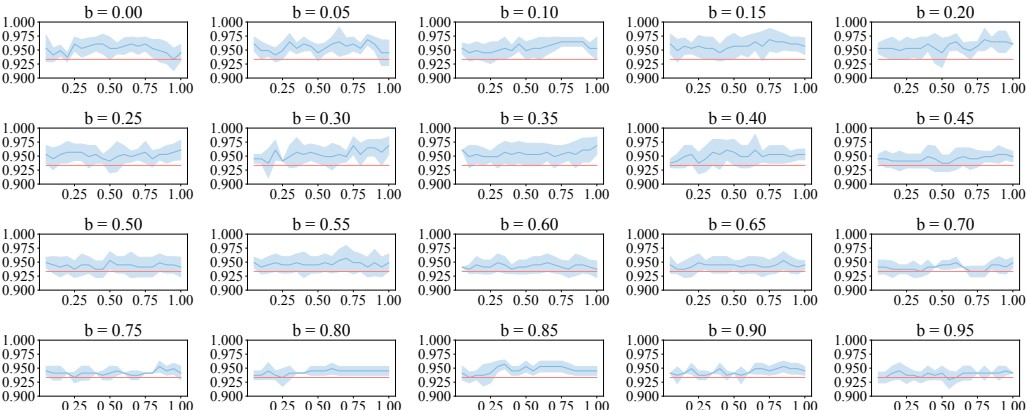

Figure 10: Acc@3 on $a$, $b$ is fixed individually.

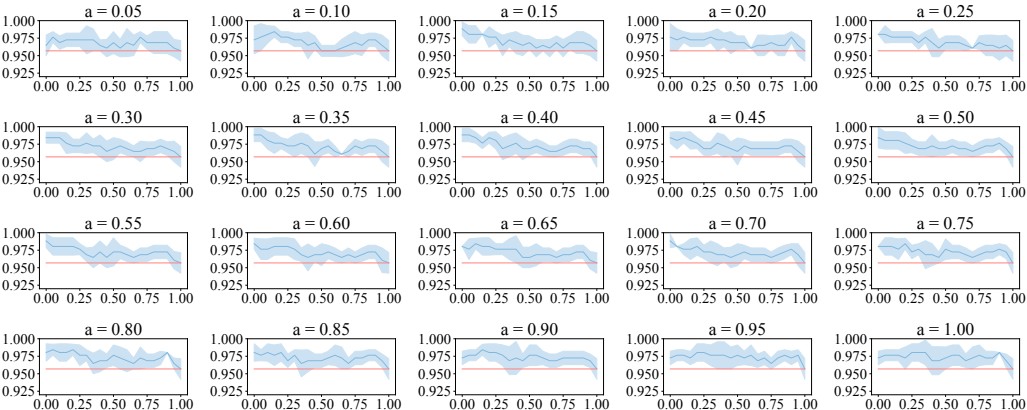

Figure 11: Acc@5 on $b$, $a$ is fixed individually.

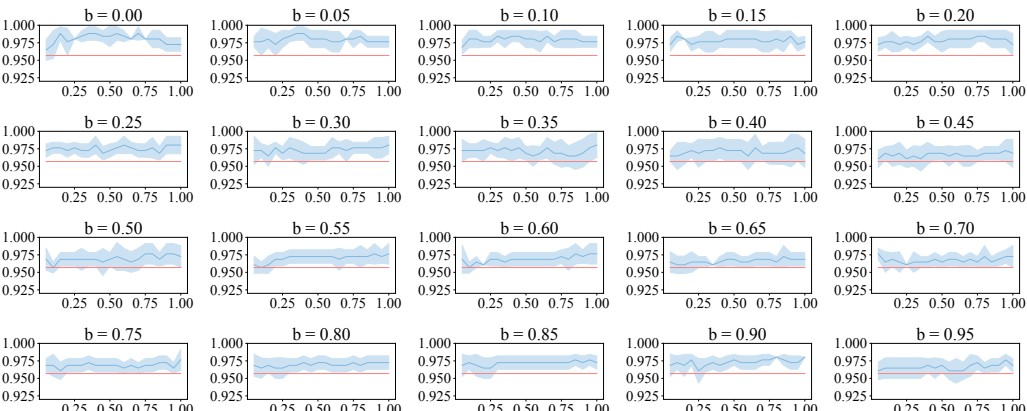

Figure 12: Acc@5 on $a$, $b$ is fixed individually.

## C.3 Combinations with More Methods

Although the mainstream of Data Free Knowledge Distillation lies in the generation methods, i.e., they rely on a generator to generate teachers' training data for compensation (Li et al., 2023), there exist some sampling methods relying on the tremendous unlabeled data samples in the wild. For example, DFND (Chen et al., 2021) identifies images most relevant to the given teacher and tasks from a large unlabeled dataset, selects useful data samples and finally uses them to conduct supervised learning to the student network with the labels the teacher give. ODSD (Wang et al., 2023b) sample open-world data close to the original data's distribution by an adaptive sampling module, introduces a low-noise representation to alleviate the domain shifts and builds a structured relationship of multiple data examples to exploit data knowledge.

When discussing the problem of OOD-KD, some readers might come up with Source-Free Domain Adaptation (Huang et al., 2021; Pei et al., 2023; Ding et al., 2022), a specific setting in Domain Adaptation. Although it falls beyond the scope of our current work, for the sake of rigor, we now highlight the differences between OOD-KD and SFDA.

SFDA assumes the absence of training data for the source models, which is akin to the scenario of the teacher model in OOD-KD. However, SFDA does its adaptation on the source model and assumes that the target model shares the same framework as the source model. The difference between source model (teacher model) and target model (student model) in the framework makes integrating teachers' knowledge into the SFDA framework remains an open problem. Moreover, some SFDA methods involve specific modifications to the backbone model, which **violates the immutability of the teacher model in OOD-KD**. For example, approaches like SHOT (Liang et al., 2020) and SHOT++ (Liang et al., 2022) divide the backbone model into feature extractor and classifier, sharing the classifier across domains. SFDA methods like C-SFDA (Karim et al., 2023) utilize confident examples for better performance. Their performance is limited when deploying to resource-constrained edge devices. What's worse, some SFDA methods base their methodology only on ResNet series models (Yang et al., 2021; Kundu et al., 2022), which is inapplicable to most lightweight neural networks.

Table 7: Results of SFDA, DFKD methods, and our proposed method. * means additional distillation progress is applied to this method.

| Office-31: Amazon, Webcam → DSLR | | | |
|---|---|---|---|
| **Method** | **Acc** | **Acc@3** | **Acc@5** |
| DFQ+ | 80.4±5.7 | 93.3±4.1 | 96.4±2.1 |
| CMI+ | 67.1±3.5 | 86.6±4.3 | 92.9±3.0 |
| DeepInv+ | 65.9±6.3 | 84.7±4.9 | 90.6±3.8 |
| w/o KD | 63.5±7.9 | 84.7±4.5 | 90.2±3.7 |
| ZSKT+ | 33.3±5.9 | 55.3±11.8 | 65.9±11.5 |
| PRE-DFKD+ | 68.3±19.5 | 87.8±14.3 | 91.8±13.3 |
| DFND+ | 59.6±7.2 | 78.4±9.6 | 88.3±4.2 |
| C-SFDA* | 62.7±4.8 | 80.8±7.0 | 89.4±6.3 |
| SFDA-DE* | 59.6±11.7 | 83.9±4.5 | 89.7±2.1 |
| U-SFDA* | 61.6±6.9 | 81.6±4.5 | 90.6±3.8 |
| **Ours** | **84.3±3.1** | **94.9±2.6** | **97.6±0.8** |

We demonstrate the results of some splendid SFDA methods (C-SFDA (Karim et al., 2023), SFDA-DE (Ding et al., 2022)), Uncertainty-SFDA (U-SFDA) (Roy et al., 2022) under the setting Office-31 Amazon, Webcam → DSLR in Table 7. Since in OOD-KD, $T$ remains immutable, adaptation is adopted to $S$ directly. As SFDA methods do not make use of ground truth labels or teachers' knowledge, in order to align with our methods, we apply additional distillation progress after employing them. However, $S$ suffers great performance degradation confronted with domain shift, similar to that observation of $T$ in main experiments. Consequently, they cannot be fully exploited, resulting in inferior performance compared to DFKD methods.

