# OpenReview forum: "AuG-KD: Anchor-Based Mixup Generation for Out-of-Domain Knowledge Distillation"
_ICLR.cc/2024/Conference — ICLR 2024 poster_

### Official Review · Reviewer_8qr9 · 2023-10-28

**Soundness:** 3 good
**Presentation:** 4 excellent
**Contribution:** 4 excellent
**Rating:** 6
**Confidence:** 3

**Summary:**

This work proposed a new task of Out-of-Domain Knowledge Distillation (OOD-KD) extended from data-free knowledge distillation, which focused on the distribution shift between teacher domain and student domain. The main challenges are (1) the absence of teacher domain's data, (2) how to selectively transfer teachers' knowledge due to the distribution shift, and (3) how to balance OOD KD and domain-specific information learning. To tackle these three challenges, the method consists of a data-free learning generator, anchor learning module, and mixup learning module. Experiments on three datasets verified the effectiveness of the proposed method.

**Strengths:**

+ This work proposed a new problem for knowledge distillation, Out-of-Domain Knowledge Distillation (OOD-KD), which is challenging and practical to solve.

+ The presentation and organization of this paper is good. It is easy to figure out the main challenges and the solutions.

+ The experimental results on Office-31, Office-Home, and VisDA-2017 reached state-of-the-art performance.

**Weaknesses:**

- The class-specific mask is essential in anchor learning, but there lacks visualizations on masks to show whether they correctly captured class-specific information.

- According to ablation study 5.3 (a), the contribution of anchor is not significant compared to mixup module, which doubts the effectiveness and necessity of anchor learning module. Here raises a question about why anchor learning is necessary and successfully selectively transfer teachers' knowledge.

- A related work [1] shared the similar motivation on the invalid IID hypothesis and the gap between teacher domain and student domain. Although the research problem is different ([1] focused on conventional KD while this work proposed OOD-KD), it would be more comprehensive if the comparison between this work and [Niu et al., NeurIPS 2022] can be discussed, especially on the above mentioned similarities on motivations, and how to selectively transfer teachers' knowledge.

Reference
[1] Respecting Transfer Gap in Knowledge Distillation. NeurIPS 2022.

**Questions:**

Please see Weaknesses.

---

> ### Author Response · Authors · 2023-11-14
> **Response to Reviewer 8qr9**
>
> First and foremost, We would like to appreciate the insightful comments and advice provided by reviewer 8qr9. Guided by these recommendations, we have made several amendments to our work, which are **highlighted in red within the PDF**.
>
> > W1: The class-specific mask is essential in anchor learning, but there lacks visualizations on masks to show whether they correctly captured class-specific information.
>
> A1: Thanks for the suggestion. According to reviewer 8qr9's advice, we respectfully add visualization results (Figure 1 and Figure 2) on the mask by t-SNE in Appendix C.1.
>
> Figure 1 (original $z$) can be referred to https://p.ipic.vip/0btpbb.png
>
> Figure 2 (masked $z$) can be referred to https://p.ipic.vip/7riqhk.png
>
> We quote it here:
>
> > **Appendix C.1  Visualization On Mask Provided by AnchorNet**
> >
> > In Module 2, AnchorNet integrates the mask operator $m$ and the mapping function $\psi$ into a lightweight neural network. The mask is class-specific and **plays a crucial role in retaining domain-invariant knowledge in the latent space**. To vividly demonstrate the effectiveness of the mask, we conduct t-SNE on the latent variables and the masked version of them. To be specific, we use AnchorNet and Encoder trained under the setting AW$\rightarrow$D in Office-31 and select 32 images for each class (31 classes in total). For each image $(x,y)$, we encode it to get the latent variable $z=E(x;\theta_e)$, and obtain the class-specific mask $m(y;\theta_a)$. Next, we obtain the masked latent variable $z'=(1-m(y;\theta_a))\odot z$. The t-SNE results on $z$ and $z'$ are displayed in Figure 1 and 2 Each displays a distribution of data points plotted against two t-SNE components. The points are colored and shaped differently to represent different classes within the latent space.
> >
> > In Figure 1, the distribution is quite mixed, with no distinct clusters or separation between the different classes. In contrast, in Figure 2, after applying mask operation on the latent variables, there appears to be a more distinct separation between different classes. Clusters of the same shapes and colors are more evident, indicating that the mask operation has enhanced the class-specific knowledge within the latent space.
>
> > W2: According to ablation study 5.3 (a), the contribution of anchor is not significant compared to mixup module, which doubts the effectiveness and necessity of anchor learning module. Here raises a question about why anchor learning is necessary and successfully selectively transfer teachers' knowledge.
>
> A2: Thanks for the question. **AnchorNet plays a crucial role in our proposed method.** We apologize for the ambiguity caused by the title of ablation results. According to the suggestions, we **eliminate the ambiguity** and **add one more ablation setting**.
>
> In Section 5.3.1, we discuss the effectiveness of our modules. As we stated, without Module 1, the proposed method cannot work, we ignore the ablation result w/o Module 1.
>
> > Since the whole method is inapplicable without Module 1, we ignore it here.
>
> As a result, conventional ablation studies would discuss the ablation result "w/o Module 2", "w/o Module 3", and "w/o Module 2,3". However, Module 2 is an auxiliary module for Module 3, as stated:
>
> > Module 2 helps Module 3 generate mixup samples for training
>
> It is also meaningless to check the ablation result of "w/o Module 3". **Therefore, our ablation study should be conducted on "w/o Module 2" and "w/o Module 2,3", which in the text are  "w/o Anchor" and "w/o Mixup" individually.**
>
> Actually, "w/o Mixup" is NOT verifying the effectiveness of Module 3: Mixup Leaning, but is **highlighting the harm of blindly adopting DFKD methods under domain shift**. As stated in the second paragraph in Section 5.3.1:
>
> > Given that w/o Mixup is a **simple process of DFKD**, the striking performance degradation **highlights the urgent need for solutions to OOD-KD**.
>
> Therefore, to eliminate ambiguity, we change the name of the ablation results from "w/o Mixup" to "M1" and from "w/o Anchor" to "M1+M3" accordingly.
>
> To further increase the rigor of this ablation study, we **dive into Module 3 and add a new ablation setting: w/o Mixup, where the model trains solely on $D_s$ without mixup samples**:

---

> > ### Author Response · Authors · 2023-11-14
> > **Continue**
> >
> > > Framework ablation studies traditionally involve masking parts of the proposed modules for experimental purposes. Yet, it is essential to recognize: 1. **Module 1 is fundamental to our method and is non-removable**; 2. Module 2 serves to support Module 3. **There is no need to test the results only with Module 1\&2**. Hence, our investigation focuses on the outcomes absent Module 2, and absent both Module 2 and Module 3, denoted as M1+M3 and M1, respectively. Additionally, our analysis dives into Module 3, where **we omit the mixup samples to evaluate their critical role, denoted as M1+M2+M3 (w/o Mixup)**. It's worth noting that **there is no need to add one more setting w/o M2 \& Mixup here** since it makes no difference to M1+M2+M3 (w/o Mixup). Consequently, we get three distinct ablation scenarios: **M1, M1+M3, and M1+M2+M3 (w/o Mixup)**. To be specific, in M1, we directly choose $S$'s best checkpoint in Module 1 and test it on $D_s$. In M1+M2+M3 (w/o Mixup), the model trains solely on $D_s$. In M1+M3, we mask AnchorNet by equating its output $z'$ with its input $z$ and then proceed with the method.
> >
> > Here, we update the ablation results:
> >
> > | Method               | Acc                             | Acc@3                            | Acc@5                             |
> > | -------------------- | ------------------------------- | -------------------------------- | --------------------------------- |
> > | M1                   | 33.7$\pm$5.1                    | 56.1$\pm$8.1                     | 68.6$\pm$3.1                      |
> > | M1+M2+M3 (w/o Mixup) | 80.1$\pm$5.7                    | 92.3$\pm$4.1                     | 96.4$\pm$2.1                      |
> > |                      | 46.4$\uparrow\pm$0.6$\uparrow$  | 36.2$\uparrow\pm$4.0$\downarrow$ | 27.8$\uparrow\pm$1.0$\downarrow$  |
> > | M1+M3                | 83.5$\pm$4.1                    | 94.5$\pm$2.9                     | 96.1$\pm$2.0                      |
> > |                      | 3.4$\uparrow\pm$1.6$\downarrow$ | 3.2$\uparrow\pm$1.2$\downarrow$  | 0.3$\downarrow\pm$0.1$\downarrow$ |
> > | M1+M2+M3             | 84.3$\pm$3.1                    | 94.9$\pm$2.6                     | 97.6$\pm$0.8                      |
> > |                      | 0.8$\uparrow\pm$1.0$\downarrow$ | 0.4$\uparrow\pm$0.3$\downarrow$  | 1.6$\uparrow\pm$1.1$\downarrow$   |
> >
> > Here, we post the edited analysis of these results:
> >
> > > The performance improvement between M1 and M1+M2+M3 (w/o Mixup) mainly stems from the supervision of student domain data. As M1 is a simple DFKD, the striking performance gap underscores the urgent need for solutions to OOD-KD. The considerable enhancement evidences the efficacy of Module 2 and Module 3 in remedying domain shifts. The rise in average accuracy coupled with reduced variance firmly attest to the significance of each component in our method.

---

> > > ### Author Response · Authors · 2023-11-14
> > > **Continue 2**
> > >
> > > > W3: A related work [1] shared the similar motivation on the invalid IID hypothesis and the gap between teacher domain and student domain. Although the research problem is different ([1] focused on conventional KD while this work proposed OOD-KD), it would be more comprehensive if the comparison between this work and [Niu et al., NeurIPS 2022] can be discussed, especially on the above mentioned similarities on motivations, and how to selectively transfer teachers' knowledge.
> > >
> > > A3: Thanks for the recommendation! IPWD is really an interesting and solid work with an original insight into Knowledge Distillation. We've added the discussion to it in the Related Work.
> > >
> > > OOD-KD and IWPD are fundamentally different. **We aim at solving the problem of domain shift in DFKD (OOD problem), while IPWD aim at solving the imbalance of the teacher's knowledge (long-tail problem).**
> > >
> > > Here, we compare its motivation and the way of selectively transferring teachers' knowledge with our method here.
> > >
> > > 1. **Comparison between motivations**. First, we should admit that our motivations **start from the same observation**, i.e. a good teacher does not necessarily result in a good student.
> > >
> > > > However, recent studies find that the teacher’s accuracy is not a good indicator of the resultant student performance.
> > >
> > > And **here comes the major differences in our motivations.**
> > >
> > > **Our work treats the observation as the result of domain shift**. We study in the context of Data-Free Knowledge Distillation, where it is natural that we are **dark about the distribution of teachers' training data**. Hence, when we conduct KD with our own data via the teacher model, we cannot guarantee that our data is IID from teachers' training data and will encounter the problem of domain shift. The problem of domain shift misleads teacher to give wrong predictions on our own data and thus undermines the student's performance. **The greater the domain shift is, the worse the student would perform**.
> > >
> > >  In contrast, **IPWD treats the observation as the result of the neglect of the imbalanced knowledge due to transfer gap**.
> > >
> > > > Even on the same training set with the same model parameter, teachers with different temperature $\tau$ **yield different “soft label” distributions from the ground-truth ones**. This implies that human and teacher knowledge are from different domains, and there is a transfer gap that drives the “dark knowledge” transferring from teacher to student ... , the transfer gap **affects the distillation performance on the under-represented classes**
> > >
> > > To be more specific, such neglect stems from **the temperature variable**, which preserves the context information by the non-dominant part in soft labels. For example, in a certain setting, a dog image could be 0.8·dog + 0.2·wolf in soft label, implying that this image could convey some wolf-like context information. E**ven if the training data is IID and class-balanced, the context information is not necessarily balanced** due to the selection strategy of the dataset. Thus **the teacher’s knowledge could be imbalanced**. The imbalance of the teacher's knowledge leads to the poor performance of classes on the tail of teacher predictions.
> > >
> > > In conclusion, **we aim at solving the problem of domain shift in DFKD, while IPWD aim at solving the imbalance of the teacher's knowledge**.
> > >
> > > 2. **Comparison between methods of knowledge transfer.** Both of us propose the way of better knowledge transfer according to the essence of the problem.
> > >
> > >    In OOD-KD, since the teacher would **give wrong predictions** under domain shift, we mixup the student domain data to transform it into the teacher domain, which the teacher could **give more reliable predictions**. To be specific, the mixup operation is instructed under the teacher's uncertainty on its prediction of a certain input.
> > >
> > >    In contrast, in IPWD, since **some knowledge is neglected**, it **raises the weight of them** for compensation. To be more specific, IPWD adjusts the weight ratio between CrossEntropy (class-specific knowledge) and KL divergence (some part is context knowledge, easy to be neglected) to pay more attention to the knowledge in the minority.
> > >
> > > Regrettably, since we are motivated differently, we cannot compare it with our proposed method directly. However, we've added it to our Related Work respectfully.
> > >
> > > > IPWD (Niu et al., 2022) also focuses on the gap between $D_t$ and $D_s$. However, different from OOD-KD, they mainly solve the imbalance in teachers' knowledge.

---

> > > > ### Comment · Reviewer_8qr9 · 2023-11-22
> > > >
> > > > Thank the authors for the detailed rebuttal, which addressed my concerns and I will keep my rating as 6.

---

> > > > > ### Author Response · Authors · 2023-11-23
> > > > >
> > > > > Thank you for the time and effort you devoted to reviewing our ICLR submission. Your insightful comments and constructive criticism have been extremely beneficial. We deeply appreciate your dedication to advancing research in our field.
> > > > >
> > > > > Wishing you continued success and fulfillment in your professional endeavors.

---

### Official Review · Reviewer_5Sg9 · 2023-10-31

**Soundness:** 3 good
**Presentation:** 3 good
**Contribution:** 3 good
**Rating:** 6
**Confidence:** 4

**Summary:**

This work studies on a new but practical problem: Out-of-Domain Knowledge Distillation (OOD-KD). OOD-KD resembles Data-Free Knowledge Distillation (DFKD), which assumes the unavailability of teacher model’s training data except for one significant difference: teacher models’ training data and student models’ test data are not IID distributed. To tackle OOD-KD, the authors propose AuG-KD to align student-domain data with the teacher domain and leverage a generative method to progressively evolve the learning process from OOD knowledge distillation to domain-specific information learning. Experimental results demonstrate its promise of the proposed methods.

**Strengths:**

1.	The problem proposed is new but practical. With the development of ML techniques, most large-scale models are released in a black box or without access to their training data. Under this circumstance, OOD problems are unavoidable. This work focuses on this novel problem and offers a practical solution.

2.	The writing is clear and sophisticated. To best clarify the problem and its solution, this work utilizes formulae, method framework, pseudocode, flow chart and visualization.

3.	The experiments are extensive. This work conducts its experiments on 3 datasets and 8 settings in five times. Besides, 3 more ablation studies are designed to substantiate the effectiveness of each module of the method. In the appendix, 3 more baselines are taken into consideration. Quite a lot of DFKD methods pay little attention to the repeatability and stability, conducting each experiment with only one seed. This work considers more rigorously in the experiment settings.

4.	The experiment analyses are detailed and convincing. Apart from analyzing the stability and superiority of their proposed method, this work steps further. They provide a clear explanation towards the high variance of each method in the data perspective, which lacks related analysis in previous studies. Besides, they provide visualization for the mixup samples generated by their methods.

**Weaknesses:**

1.	The detail of the scheduler function is not released. Although the important characteristics of scheduler functions are provided in the main body, I cannot figure out the specific scheduler function used in the experiments.
2.	The ablation study of the effectiveness of each component is not clear. The setting seems ambiguous and in need of further explanation.

**Questions:**

1.	The same to weakness 1. To be specific, could you provide more details about which scheduler function you use and what kind of scheduler function would be preferred?

2.	The same to weakness 2. More specifically, the ablation result of w/o Anchor has a great performance improvement compared to that of w/o Mixup. Could you further explain it?

3.	This work provides detailed visualization on the mixup sample generated in the main body. We could see a gradual change as f increases. However, are there any relationships between the imgs at the same position with different f?

---

> ### Author Response · Authors · 2023-11-14
> **Response to Reviewer 5Sg9**
>
> First and foremost, We would like to appreciate the insightful comments and advice provided by reviewer 5Sg9. Guided by these recommendations, we have made several amendments to our work, which are **highlighted in red within the PDF**.
>
> > W1: The detail of the scheduler function is not released. Although the important characteristics of scheduler functions are provided in the main body, I cannot figure out the specific scheduler function used in the experiments.
>
> A1: Thanks for the question. For the sake of space, we did not release the details of the scheduler function we used in our experiments.  As stated in the main body, the scheduler function should follow 2 properties:
>
> 1. $F(a\cdot\sharp{\rm Epoch};a,b)=1$
> 2. $F(0;a,b)=b$
>
> Besides, it should be monotonically non-decreasing within its domain. In our experiments, we only use the simplest scheduler function - linear function for all settings. When $a$ and $b$ are given, the scheduler function can be determined.
>
> In fact, the idea of scheduler function draws inspiration from Curriculum Learning [1]. For the performance of different and more complex scheduler functions, we refer readers4 to [1] for further reading.
>
> The discussion of scheduler functions we use has been added to **the end of Appendix A**.
>
> > W2: The ablation study of the effectiveness of each component is not clear. The setting seems ambiguous and in need of further explanation.
>
> A2: Thanks for the advice. We apologize for the ambiguity caused by the title of ablation results. According to the suggestions, we **eliminate the ambiguity** and **add one more ablation setting**.
>
> In Section 5.3.1, we discuss the effectiveness of our modules. As we stated, without Module 1, the proposed method cannot work, we ignore the ablation result w/o Module 1.
>
> > Since the whole method is inapplicable without Module 1, we ignore it here.
>
> As a result, conventional ablation studies would discuss the ablation result "w/o Module 2", "w/o Module 3", and "w/o Module 2,3". However, Module 2 is an auxiliary module for Module 3, as stated:
>
> > Module 2 helps Module 3 generate mixup samples for training
>
> It is also meaningless to check the ablation result of "w/o Module 3". **Therefore, our ablation study should be conducted on "w/o Module 2" and "w/o Module 2,3", which in the text are  "w/o Anchor" and "w/o Mixup" individually.**
>
> Actually, "w/o Mixup" is NOT verifying the effectiveness of Module 3: Mixup Leaning, but is **highlighting the harm of blindly adopting DFKD methods under domain shift**. As stated in the second paragraph in Section 5.3.1:
>
> > Given that w/o Mixup is a **simple process of DFKD**, the striking performance degradation **highlights the urgent need for solutions to OOD-KD**.
>
> Therefore, to eliminate ambiguity, we change the name of the ablation results from "w/o Mixup" to "M1" and from "w/o Anchor" to "M1+M3" accordingly.
>
> To further increase the rigor of this ablation study, we **dive into Module 3 and add a new ablation setting: w/o Mixup, where the model trains solely on $D_s$ without mixup samples**:
>
> > Framework ablation studies traditionally involve masking parts of the proposed modules for experimental purposes. Yet, it is essential to recognize: 1. **Module 1 is fundamental to our method and is non-removable**; 2. Module 2 serves to support Module 3. **There is no need to test the results only with Module 1\&2**. Hence, our investigation focuses on the outcomes absent Module 2, and absent both Module 2 and Module 3, denoted as M1+M3 and M1, respectively. Additionally, our analysis dives into Module 3, where **we omit the mixup samples to evaluate their critical role, denoted as M1+M2+M3 (w/o Mixup)**. It's worth noting that **there is no need to add one more setting w/o M2 \& Mixup here** since it makes no difference to M1+M2+M3 (w/o Mixup). Consequently, we get three distinct ablation scenarios: **M1, M1+M3, and M1+M2+M3 (w/o Mixup)**. To be specific, in M1, we directly choose $S$'s best checkpoint in Module 1 and test it on $D_s$. In M1+M2+M3 (w/o Mixup), the model trains solely on $D_s$. In M1+M3, we mask AnchorNet by equating its output $z'$ with its input $z$ and then proceed with the method.

---

> > ### Author Response · Authors · 2023-11-14
> > **Continue**
> >
> > Here, we update the ablation results:
> >
> > | Method               | Acc                             | Acc@3                            | Acc@5                             |
> > | -------------------- | ------------------------------- | -------------------------------- | --------------------------------- |
> > | M1                   | 33.7$\pm$5.1                    | 56.1$\pm$8.1                     | 68.6$\pm$3.1                      |
> > | M1+M2+M3 (w/o Mixup) | 80.1$\pm$5.7                    | 92.3$\pm$4.1                     | 96.4$\pm$2.1                      |
> > |                      | 46.4$\uparrow\pm$0.6$\uparrow$  | 36.2$\uparrow\pm$4.0$\downarrow$ | 27.8$\uparrow\pm$1.0$\downarrow$  |
> > | M1+M3                | 83.5$\pm$4.1                    | 94.5$\pm$2.9                     | 96.1$\pm$2.0                      |
> > |                      | 3.4$\uparrow\pm$1.6$\downarrow$ | 3.2$\uparrow\pm$1.2$\downarrow$  | 0.3$\downarrow\pm$0.1$\downarrow$ |
> > | M1+M2+M3             | 84.3$\pm$3.1                    | 94.9$\pm$2.6                     | 97.6$\pm$0.8                      |
> > |                      | 0.8$\uparrow\pm$1.0$\downarrow$ | 0.4$\uparrow\pm$0.3$\downarrow$  | 1.6$\uparrow\pm$1.1$\downarrow$   |
> >
> > Here, we post the edited analysis of these results:
> >
> > > The performance improvement between M1 and M1+M2+M3 (w/o Mixup) mainly stems from the supervision of student domain data. As M1 is a simple DFKD, the striking performance gap underscores the urgent need for solutions to OOD-KD. The considerable enhancement evidences the efficacy of Module 2 and Module 3 in remedying domain shifts. The rise in average accuracy coupled with reduced variance firmly attest to the significance of each component in our method.
> >
> > > Q3: This work provides detailed visualization on the mixup sample generated in the main body. We could see a gradual change as f increases. However, are there any relationships between the imgs at the same position with different f?
> >
> > A3: Thanks for the question. Figure 3 provides Different mixup samples generated in Module 3 for DSLR in Office-31, controlled by the stage factor $f \in[0, 1]$. There ARE relationships between the imgs at the same position with different $f$. **The images at the same position with different $f$ stem from the same image in the student domain, i.e. $f=1$. The only difference between the images in the same position is the value $f$, which controls the image by Equation 11.**
> >
> > [1]  Dynamic curriculum learning for imbalanced data classification, ICCV 2019

---

> > > ### Comment · Reviewer_5Sg9 · 2023-11-23
> > >
> > > Thanks for the detailed response. I have read the rebuttal and most of my concerns have been well addressed. Overall, I am towards acceptance and will maintain my score.
> > >
> > > Best,
> > > The reviewer 5Sg9

---

> > > > ### Author Response · Authors · 2023-11-23
> > > >
> > > > Thank you for your constructive comments on our work! Your expertise has been incredibly helpful.
> > > >
> > > > Best wishes for your ongoing work.

---

### Official Review · Reviewer_hjEr · 2023-10-31

**Soundness:** 1 poor
**Presentation:** 2 fair
**Contribution:** 2 fair
**Rating:** 6
**Confidence:** 3

**Summary:**

This paper explores a significant and practical problem, Out-of-Domain Knowledge Distillation (OOD-KD). The authors believe that adopting models derived from DFKD for real-world applications suffers significant performance degradation, due to the discrepancy between teachers’ training data and real-world scenarios (student domain). Therefore, teachers’ knowledge must be selectively transferred. So the authors proposed AuG-KD, which utilizes an uncertainty-guided and sample-specific anchor to align student-domain data with the teacher domain. Experiments illustrate the effectiveness of the method.

**Strengths:**

1. It is valuable to explore the relationship between OOD and data-free distillation.

2. Although experiments were conducted on multiple dataset settings, the scale was small.

**Weaknesses:**

1. The motivation behind the setting of this paper is confusing. Why do we need to use out-of-distribution teachers when we have student domain data? Wouldn’t it be better to directly train the teacher under the student domain?
2. Lack of discussion with DFND[1], MosiacKD[2] and ODSD[3]. We think that since generation methods are known, sampling methods should also be compared.
3. Why not conduct experiments with CIFAR10 and CIFAR100? This is the mainstream dataset in the DFKD. Furthermore, can this method generalize on ImageNet?

[1] Learning Student Networks in the Wild, CVPR 2021

[2] Mosaicking to Distill: Knowledge Distillation from Out-of-Domain Data, NIPS 2021

[3] Sampling to Distill: Knowledge Transfer from Open-World Data, arxiv 2023

**Questions:**

See Weaknesses.

---

> ### Author Response · Authors · 2023-11-14
> **Response to Reviewer hjEr**
>
> First and foremost, We would like to appreciate the insightful comments and advice provided by reviewer hjEr. Guided by these recommendations, we have made several amendments to our work, which are **highlighted in red within the PDF**.
>
> > W1: The motivation behind the setting of this paper is confusing. Why do we need to use out-of-distribution teachers when we have student domain data? Wouldn’t it be better to directly train the teacher under the student domain?
>
> A1: Thanks for the question.
>
> First, **the problem of OOD-KD is ubiquitous**. For practical applications, **deploying large models directly onto users' devices**, such as using facial recognition at entry points or object detection in autonomous driving scenarios, often **isn't feasible** due to their size and resource requirements. Instead, we deploy a lightweight neural network (e.g. MobileNet) and train it with  i ts data and our large-scale teacher model. However, **their data are naturally OOD with each other and of course, the teachers' training data** (e.g. ethnicity, skin tone in facial recognition and operating environment and weather condition in autonomous) .
>
> In this case, directly distilling the teacher into our model would lead to great performance degradation. To further improve our model's performance, we should address OOD-KD.
>
> * The reason why we need to use out-of-distribution teachers when we have student domain data is that **directly training the student under the student domain is  not good enough, especially when the amount of data is not sufficient**.
>
>   In our main experiments, we have a baseline "w/o KD":
>
> > One more baseline "w/o KD" is to train the student model S without the assistance of $T$, starting with weights pre-trained on ImageNet (Deng et al., 2009).
>
> "w/o KD" trains a model solely with student domain data, which is not good enough compared to other methods.
>
> Due to space limitations, We put part of the experiment results here. For more detailed results, we kindly refer readers to our original submission. In Office-31,
>
> | Setting | A,W$\rightarrow$D |              |              | A,D$\rightarrow$W |              |              | D,W$\rightarrow$A |              |              |
> | ------- | ----------------- | ------------ | ------------ | ----------------- | ------------ | ------------ | ----------------- | ------------ | ------------ |
> | Metric  | Acc               | Acc@3        | Acc@5        | Acc               | Acc@3        | Acc@5        | Acc               | Acc@3        | Acc@5        |
> | w/o KD  | 63.5$\pm$7.9      | 84.7$\pm$4.5 | 90.2$\pm$3.7 | 82.7$\pm$5.4      | 96.0$\pm$1.9 | 98.3$\pm$0.7 | 52.9$\pm$3.4      | 72.5$\pm$3.6 | 79.9$\pm$2.2 |
> | Ours    | 84.3$\pm$3.1      | 94.9$\pm$2.6 | 97.6$\pm$0.8 | 87.8$\pm$7.6      | 96.3$\pm$1.8 | 99.5$\pm$0.7 | 58.8$\pm$3.7      | 73.7$\pm$2.1 | 79.7$\pm$1.5 |
>
> The performance gap between "w/o KD" and our proposed method is nearly 5% in VisDA-2017, 4% in Office-Home, and 20% in Office-31.  The performance gap is larger and larger as the scale of student domain data decreases but is still significant with 280000 images (VisDA-2017).  **The performance gap underlines the importance of applying KD for better student-domain performance**, especially when the student domain data is not sufficient.
>
> * Usually,  **we CANNOT directly train the teacher under the student domain** for 2 main reasons.
>   1. **We may not be able to deploy a large-scale teacher model on our devices** like mobile phones, etc, due to resource limitations.
>   2. **Some large models are immutable to us due to patent restrictions or resource limitations, like GPT-4.**  It is NOT practical for us to finetune them for better performance.
>
> In conclusion, addressing the problem of OOD-KD is **of great importance**, since it enables us to transfer knowledge from any teacher model we want and improve the performance of our own models.

---

> > ### Author Response · Authors · 2023-11-14
> > **Continue**
> >
> > > W2: Lack of discussion with DFND[1], MosiacKD[2] and ODSD[3]. We think that since generation methods are known, sampling methods should also be compared.
> >
> > A2: Thanks for the advice. Besides the generation methods in DFKD, sampling methods are also an interesting branch.We have addd discussion with them in **the Related Work** and **detailed them in the Appendix C.3**.
> >
> > Here, we quote the discussions here.
> >
> > > Although the mainstream of Data Free Knowledge Distillation lies in the generation methods, i.e., they rely on a generator to generate teachers' training data for compensation, there exist some sampling methods relying on the tremendous unlabeled data samples in the wild.
> >
> > 1. As to DFND, we add it to the baselines and the results are displayed in Appendix C.3
> >
> >    > DFND (Chen et al., 2021) identifies images most relevant to the given teacher and tasks from a large unlabeled dataset, selects useful data samples and finally uses them to conduct supervised learning to the student network with the labels the teacher give.
> >
> >    We also display its result here:
> >
> >    | Method | Acc          | Acc@3        | Acc@5        |
> >    | ------ | ------------ | ------------ | ------------ |
> >    | DFND+  | 59.6$\pm$7.2 | 78.4$\pm$9.6 | 88.3$\pm$4.2 |
> >    | Ours   | 84.3$\pm$3.1 | 94.9$\pm$2.6 | 97.6$\pm$0.8 |
> >
> > 2. As to MosaicKD, we've already discussed it in the **Related Work**. MosaicKD  aim at utilizing out-of-domain data for better in-domain performance. Since our motivation are different, we do not list MosaicKD as a baseline but appreciate their explorations.
> >
> >    > MosiacKD first proposes the concept of Out-of-Domain Knowledge Distillation but their objective is **fundamentally different from ours**. They use OOD data to assist source-data-free knowledge distillation and focus on **in-domain performance** . In contrast, we use OOD data for better **out-of-domain performance**.
> >
> >    According to the valuable advice, we also display its result here:
> >
> >    | Method    | Acc           | Acc@3        | Acc@5        |
> >    | --------- | ------------- | ------------ | ------------ |
> >    | MosaicKD+ | 60.8$\pm$10.8 | 80.0$\pm$6.3 | 85.5$\pm$4.1 |
> >    | Ours      | 84.3$\pm$3.1  | 94.9$\pm$2.6 | 97.6$\pm$0.8 |
> >
> > 3. As to ODSD,  we discuss it in the Appendix C.3.
> >
> >    > ODSD (Wang et al., 2023b) sample open-world data close to the original data’s distribution by an adaptive sampling module, introduces a low-noise representation to alleviate the domain shifts and builds a structured relationship of multiple data examples to exploit data knowledge.
> >
> > However, as we have emphasized in our discussion, **DFKD methods CANNOT solve the problem of OOD-KD since they neglect the domain shift problem between teachers' training data and student domain data.** Actually, all the DFKD baselines in our experiments have an additional finetune process afterwards:
> >
> > > Given that OOD-KD is a relatively novel problem, there are no readily available baselines. Instead, we adopt state-of-the-art DFKD methods, ... and fine-tune them on the student domain.
> >
> > > W3: Why not conduct experiments with CIFAR10 and CIFAR100? This is the mainstream dataset in the DFKD. Furthermore, can this method generalize on ImageNet?
> >
> > A3: Thanks for the question.
> >
> > 1. Since we are discussing the problem of OOD-KD and are focusing on the OOD performance, it is important to choose datasets with several domains. Office-31, Office-Home, and VisDA-2017 are the top-3 commonly used datasets in the context of domain adaptation. So, we follow the setting in DA and choose these 3 datasets for experiments.
> > 2. Although CIFAR10 and CIFAR100 are commonly used in DFKD, they are not divided into several domains. For the sake of rigor, we can not conduct experiments on them.
> > 3. Our method **can** generalize to ImageNet **IF it offers data from different domains.** Like CIFAR10 and CIFAR100, ImageNet does not have data in different domains. With respect to the effectiveness of our methods on large-scale datasets, in our experiments, VisDA-2017 is a large-scale dataset containing 280000 images. The results on VisDA-2017 demonstrate the effectiveness of our method on large-scale datasets.

---

> > > ### Comment · Reviewer_hjEr · 2023-11-22
> > >
> > > The response addresses our main concerns, so we increase the rating.

---

> > > > ### Author Response · Authors · 2023-11-22
> > > >
> > > > Thank you for your valuable feedback on our submission. Your suggestions have significantly improved our work! We appreciate your dedication and expertise.
> > > >
> > > > Wishing you all the best in your future endeavors.

---

### Official Review · Reviewer_aobZ · 2023-11-01

**Soundness:** 2 fair
**Presentation:** 3 good
**Contribution:** 2 fair
**Rating:** 6
**Confidence:** 4

**Summary:**

The study introduces a technique for knowledge distillation wherein a student model learns from a pre-trained teacher model, even when the domains differ. Assuming a consistent label space across both domains, the research addresses the inherent distribution shift. It suggests the development of an 'anchor network' designed to extract domain-neutral features within the latent space. These domain-agnostic images, generated via latent features, are then employed as samples, mixed with student domain data, to facilitate efficacious distillation.

**Strengths:**

1. The manuscript is articulated with clarity, ensuring effective comprehensibility to the readers.

2. The authors masterfully contextualize their work in the introduction, clearly delineating their approach from conventional Generic KD and DFKD paradigms.

3. Their introduction of the anchor-based mixup strategy not only showcases novelty but also delivers a marked improvement in knowledge distillation outcomes when benchmarked against prior DFKD techniques and selected source-free domain adaptation methods.

4. The strategy to employ a mask for distinguishing between domain-specific and domain-neutral features is underpinned by a solid rationale, and its formulation and application are both commendably executed.

**Weaknesses:**

Major:

1. The proposed problem definition appears to closely resemble source-free domain adaptation (SFDA) [1]. The only distinction being that the target domain network is more lightweight compared to the pre-trained teacher network. As a result, the problem setup seems to simply be a specific instance of the broader source-free domain adaptation scenario. This raises concerns about the novelty of the problem statement.

2. Given the dataset $D_{s}$, which consists of OOD student domain data with labels, I wonder about the performance of the student when trained solely on $D_{s}$  in a supervised manner. Did the authors conduct any initial tests to gauge baseline performance? Without such a baseline, I question how the authors determined the severity of the problem.

Minor:

1. In Equation 2, the left-hand side (LHS) includes $z_{0}$  in its argument, but $z_{0}$ is absent from the right-hand side (RHS). Clarification is needed regarding the functional operations. This inconsistency is observed in Equations 2 and 3 as well.

2. For Equation 4, while the expectation is based on $z_{0}$, $z_{0}$ is not reflected in the RHS's loss combination. A similar issue is present in Equation 6.

3. In Equation 4, the loss function $L_{generator}$ is introduced. However, it's unclear whether this loss is optimizing the generator weights, the latent space, or both. This lack of clarity persists in Equations 5, 6, and 10.

[1] Kurmi, Vinod K., Venkatesh K. Subramanian, and Vinay P. Namboodiri. "Domain impression: A source data free domain adaptation method." Proceedings of the IEEE/CVF winter conference on applications of computer vision. 2021.

**Questions:**

See Weaknesses

---

> ### Author Response · Authors · 2023-11-14
> **Response to Reviewer aobZ**
>
> First and foremost, We would like to appreciate the insightful comments and advice provided by reviewer aobZ. Guided by these recommendations, we have made several amendments to our work, which are **highlighted in red within the PDF**.
>
> > Major 1: The proposed problem definition appears to closely resemble source-free domain adaptation (SFDA) [cite]. The only distinction being that the target domain network is more lightweight compared to the pre-trained teacher network. As a result, the problem setup seems to simply be a specific instance of the broader source-free domain adaptation scenario. This raises concerns about the novelty of the problem statement.
>
> A1: Thanks for the suggestion.  It is of great importance to distinguish OOD-KD from SFDA. We value this input and believe readers will find a more comprehensive exploration of this topic in **Appendix C.3**. I will quote some discussions below.
>
> To clarify this concern, it is important to note that SFDA and OOD-KD are **fundamentally different in 2 aspects**:
>
> 1. In OOD-KD, source and target model are **architecture-agnostic**.
>
>    To be specific,  in most scenarios, we need a lightweight model suitable for our data. Thus we train it with our data and improve its performance with the assistance of a large-scale teacher model. This is the common scenario of KD, where the source model (teacher) and the target model (student) are **naturally different in architecture**. However, **the inaccessibility of teachers’ training data often leads to the problem of OOD-KD and we should address the domain shift problem (difference in distribution between teachers' training data and our data) for better performance**.
>
>    On the other hand, SFDA aims at **adjusting current model to fit a new domain, i.e, adapt source model to target domain**, where target model and source model share the same architecture and **base their methods on this assumption**.
>
>    **The difference in motivation renders SFDA methods unsuitable for OOD-KD, as they operate under the assumption of architecture identity**. Therefore, even we have splendid DFDA methods, we need to address the problem of OOD-KD.
>
> 2. In OOD-KD, source model is **immutable**. To be specific, the immutability of large-scale teacher model is quite common in real-world settings due to privacy or resource limitations. For example, it is not practical for us to finetune models like GPT-4.
>
> These 2 properties are natural in the context of knowledge distillation but is usually neglected in SFDA methods. For example, SDDA[1]  (reviewer aobZ mentioned) consists of a Domain Adaptation Module, which **consists of a shared feature extractor for source and target domain datasets**. We also discuss some other SFDA methods in the **last second paragraph of Appendix C.3**.
>
> > **However, SFDA does its adaptation on the source model and assumes that the target model shares the same framework as the source model.** ... For example, approaches like SHOT (Liang et al., 2020) and SHOT++ (Liang et al., 2022) divide the backbone model into feature extractor and classifier, sharing the classifier across domains. ...... What’s worse, some SFDA methods base their methodology only on ResNet series models (Yang et al., 2021; Kundu et al., 2022), which is inapplicable to most lightweight neural networks.
> > Major 2: Given the dataset $D_s$, which consists of OOD student domain data with labels, I wonder about the performance of the student when trained solely on $D_s$ in a supervised manner. Did the authors conduct any initial tests to gauge baseline performance? Without such a baseline, I question how the authors determined the severity of the problem.
>
> A2: Thanks for the suggestions. **Yes, we conduct some tests to gauge baseline performance.** The baseline of training solely on $D_s$ is necessary to demonstrate the urgent need for practical solutions to OOD-KD. We appreciate the insight on this matter, which has been detailed within the main text of our article, which is **the baseline w/o KD** in the initial submission:
>
> > One more baseline "w/o KD" is to train the student model S without the assistance of $T$, starting with weights pre-trained on ImageNet (Deng et al., 2009).
>
> As stated in the **last second paragraph in Section 5.1**, "w/o KD" adopts the initialization pre-trained on ImageNet and is solely trained with $D_s$ without the help of teacher model. We also put part of the results here.

---

> > ### Author Response · Authors · 2023-11-14
> > **Continue**
> >
> > In Office-31,
> >
> > | Setting | A,W$\rightarrow$D |              |              | A,D$\rightarrow$W |              |              | D,W$\rightarrow$A |              |              |
> > | ------- | ----------------- | ------------ | ------------ | ----------------- | ------------ | ------------ | ----------------- | ------------ | ------------ |
> > | Metric  | Acc               | Acc@3        | Acc@5        | Acc               | Acc@3        | Acc@5        | Acc               | Acc@3        | Acc@5        |
> > | w/o KD  | 63.5$\pm$7.9      | 84.7$\pm$4.5 | 90.2$\pm$3.7 | 82.7$\pm$5.4      | 96.0$\pm$1.9 | 98.3$\pm$0.7 | 52.9$\pm$3.4      | 72.5$\pm$3.6 | 79.9$\pm$2.2 |
> > | Ours    | 84.3$\pm$3.1      | 94.9$\pm$2.6 | 97.6$\pm$0.8 | 87.8$\pm$7.6      | 96.3$\pm$1.8 | 99.5$\pm$0.7 | 58.8$\pm$3.7      | 73.7$\pm$2.1 | 79.7$\pm$1.5 |
> >
> > From the results, we can conclude that training solely on $D_s$ is not good enough, thus implying the severity of OOD-KD.
> >
> > > Minor 1: In Equation 2, the left-hand side (LHS) includes $z_0$ in its argument, but  $z_0$ is absent from the right-hand side (RHS). Clarification is needed regarding the functional operations. This inconsistency is observed in Equations 2 and 3 as well.
> >
> > A3: Thanks for the advice. We leave some variables declared within the text for more space. $z_0$ is absent in RHS in Equations 2, 3, 4, and 6, because we use $x$ and $\tilde{x}$ derived from $z_0$ here. $x$ and $\tilde{x}$ are defined in **the first paragraph of Section 4.1**:
> >
> > > The generated image is denoted as $x=G(z_0; \theta_g)$, while the normalized version of it is $ \tilde{x}= N(x)$
> >
> > According to your suggestions, we bold these statements in the new revision.
> >
> > > Minor 2: For Equation 4, while the expectation is based on $z_0$, $z_0$ is not reflected in the RHS's loss combination. A similar issue is present in Equation 6.
> >
> > A4: Thanks for the advice. We leave some variables declared within the text for more space. $z_0$ is absent in RHS in Equations 2, 3, 4, and 6, because we use $x$ and $\tilde{x}$ derived from $z_0$ here. $x$ and $\tilde{x}$ are defined in **the first paragraph of Section 4.1**:
> >
> > > The generated image is denoted as $x=G(z_0; \theta_g)$, while the normalized version of it is $ \tilde{x}= N(x)$
> >
> > According to your suggestions, we bold these statements in the new revision.
> >
> > > Minor 3: In Equation 4, the loss function $L_{\rm generator}$ is introduced. However, it's unclear whether this loss is optimizing the generator weights, the latent space, or both. This lack of clarity persists in Equations 5, 6, and 10.
> >
> > A5: Thanks for the advice. We will answer the question one by one.
> >
> > 1.  As to Equation 4: $L_\rm{generator}$, it trains the weights of the generator. The latent variable $z_0$ samples from a normal distribution $\mathcal{N}(0,1)$
> > 2. As to Equation 5, it trains the encoder weights. The encoder is trained with the generator fixed with $L_\rm{Encoder}$. In one epoch of training, we **first optimize the generator weights** and then **fix the generator and train encoder and student by Equation 5 & 6**.
> > 3. As to Equation 6, it trains the student weights. The student is trained together with the encoder when we fix the parameter of the generator. We've provided detailed training pseudo-codes in **Appendix A**:
> >
> > > for $i\gets1$ to $\sharp\rm{epoch}$ do
> > >
> > > ​	Sample $z_0$ from normal distribution with size $(b, N_z)$
> > >
> > > ​	Compute $L_\rm{generator}$ and update $\theta_g$
> > >
> > > ​	for $j\gets1$ to 5 do
> > >
> > > ​		Compute $L_\rm{encoder}$, $L_\rm{student}$ and update $\theta_e$, $\theta_t$
> > >
> > > ​	end
> > >
> > > ​	...
> > >
> > > end
> >
> > To eliminate ambiguity, we've edited **the second paragraph in Section 4.1**:
> >
> > > **Meanwhile**, an **additional **encoder $E(\cdot;\theta_e): X,Y\mapsto{Z}$ is trained**, keeping $\theta_g$ fixed**.
> > >
> > > **When training the encoder, t**he student model $S$ is trained simultaneously with Eq. 6.
> >
> > 4. As to Equation 10, it trains the AnchorNet weights. The training process of AnchorNet is independent of those of the generator, encoder, and student. It takes student domain data (sample from $P_s$) as input, computes the loss and finally optimize AnchorNet weight $\theta_a$.
> >
> > [1] Kurmi, Vinod K., Venkatesh K. Subramanian, and Vinay P. Namboodiri. "Domain impression: A source data free domain adaptation method." Proceedings of the IEEE/CVF winter conference on applications of computer vision. 2021.

---

> > > ### Comment · Reviewer_aobZ · 2023-11-22
> > >
> > > Thank you for the comprehensive rebuttal. It addresses most of my concerns, as well as those raised by other reviewers. As a result, I have decided to increase my rating for your submission.
> > >
> > > However, I would like to recommend an additional enhancement for clarity's sake. Given the multifaceted nature of the tasks in your method, it would be highly beneficial if the equations in your manuscript could be made more self-explanatory. Specifically, it would help readers to understand which parameters are being optimized by which objectives more clearly. This modification would significantly aid in comprehending the intricacies of your approach, especially considering the complexity arising from the multiple tasks being performed.
> > >
> > > Thank you once again for your attention to these details.

---

> > > > ### Author Response · Authors · 2023-11-23
> > > >
> > > > Thanks again for your constructive suggestions. We have made a new revision of our submission for more clarity.
> > > >
> > > > To be specific, **we have explicitly signified which parameters are trained in the losses.**
> > > >
> > > > As to $L_{\rm generator}$, we add a new line
> > > > > $\hat{\theta_g}=\arg\min_{\theta_g} L_{\rm generator}$
> > > >
> > > > As to $L_{\rm encoder}$, we change the equation to
> > > > > $\hat{\theta_e}=\arg\min_{\theta_e}L_{\rm encoder}=\arg\min_{\theta_e}\mathbb{E}_{z_0\sim \mathcal{N}(0,1)}\Bigl[{\rm MSE}(z_0,z)+\alpha_e\cdot{\rm KL}(z\parallel z_0)\Bigl]$
> > > >
> > > > **$L_{\rm student}$ and $L_{\rm anchor}$ are also changed in the similar way.**
> > > >
> > > > All in all, we are grateful for your guidance and the high standards you uphold in the review process. Your contribution to our work is immensely appreciated. Wishing you all the best in your continued academic and professional journey.

---

### Meta-Review · Area_Chair_Dojh · 2023-12-10

**Metareview:**

The authors address the challenge of out-of-domain (OOD) knowledge distillation which is very critical when the training data of teacher model are not accessible. A method named AuG-KD is proposed to employ a data-driven anchor to align student-domain data with the teacher domain, using a generative method to gradually shift from OOD knowledge distillation to domain-specific information learning. During the rebuttal, the authors included additional experimental results (e.g., additional baselines such as DFND, MosaicKD and ODSD) and clarifications. Overall, the reviewers were satisfied with the responses from the authors, and all of them agreed on a weak accept for this paper.

**Justification For Why Not Higher Score:**

I agree with the reviewers that the problem setting is new, the proposed solution is practical and the evaluation is quite comprehensive and convincing. Therefore, I would like to recommend accepting this paper. Given that no reviewers gave a score >= 8, I would prefer not to recommend an Accept with spotlight.

**Justification For Why Not Lower Score:**

All the reviewers agreed on a weak accept for this paper and thus I am also inclined towards an accept, instead of reject.

---

### Decision · Program_Chairs · 2024-01-16

Accept (poster)